# A Bayesian Approach for Personalized Federated Learning in Heterogeneous Settings

**Disha Makhija**
Electrical and Computer Engineering
University of Texas at Austin
Austin, TX 78705
disham@utexas.edu

**Joydeep Ghosh**
Electrical and Computer Engineering
University of Texas at Austin
Austin, TX 78705
jghosh@utexas.edu

**Nhat Ho**
Statistics and Data Science
University of Texas at Austin
Austin, TX 78705
minhnhat@utexas.edu

## Abstract

Federated learning (FL), through its privacy-preserving collaborative learning approach, has significantly empowered decentralized devices. However, constraints in either data and/or computational resources among participating clients introduce several challenges in learning, including the inability to train large model architectures, heightened risks of overfitting, and more. In this work, we present a novel FL framework grounded in Bayesian learning to address these challenges. Our approach involves training personalized Bayesian models at each client tailored to the unique complexities of the clients' datasets and efficiently collaborating across these clients. By leveraging Bayesian neural networks and their uncertainty quantification capabilities, our local training procedure robustly learns from small datasets. And the novel collaboration procedure utilizing priors in the functional (output) space of the networks facilitates collaboration across models of varying sizes, enabling the framework to adapt well in heterogeneous data and computational settings. Furthermore, we present a differentially private version of the algorithm, accompanied by formal differential privacy guarantees that apply without any assumptions on the learning algorithm. Through experiments on popular FL datasets, we demonstrate that our approach outperforms strong baselines in both homogeneous and heterogeneous settings, and under strict privacy constraints.

## 1 Introduction

Federated Learning (FL) has emerged as a pivotal paradigm in various real-world applications, offering a decentralized approach that allows participating clients to contribute to a shared model without compromising the privacy of their raw data. However, implementing FL in practical scenarios poses challenges due to the significant variability among participating clients in terms of their local data and computational resources. Clients with restricted compute capacity may encounter difficulty in training large machine learning models identical to those of other clients, and those with minimal data may struggle to obtain reliable estimates of local model parameters.

Due to their ability to generalize under limited data and provide uncertainty quantifications [69, 2, 3], we consider Bayesian based learning methods to construct improved local models. Employing Bayesian learning in FL, however, would involve the following steps - each client doing local

posterior inference to obtain a distribution over weight parameters and then communicating the local posteriors to the server, the server receiving the local posteriors from the clients and aggregating them to obtain a global posterior distribution, which is then broadcast to the clients for the next round of training. This entire learning procedure, as it turns out, is highly resource and communication-intensive. For solving an $m$-dimensional federated least squares estimation, this method will require $O(m^3)$ computation on all the clients and server sites [4] which is much more as opposed to the cost of standard FL (generally $O(m)$). *How could we then utilize the strengths of Bayesian methods for FL settings without paying such high costs?* Additionally, *given substantial variations in locally available computing resources, how do we still enable efficient learning and collaboration on all clients?* We address these questions by proposing a framework that allows all clients to train their own personal Bayesian models (with varying model complexities), and achieves collaboration across clients by distilling knowledge from the peer clients via a shared unlabelled public dataset and instilling that knowledge in the local models in the form of priors. The challenge of transferring knowledge across models of different architectures is addressed by using the functional (output) space to indirectly determine priors on weights of the local model parameters. Furthermore, to prevent the data leaks in FL setup [22, 68, 23] and formally guarantee the privacy of the local client data, we present a differentially-private version of the algorithm applying a formal well-known standard of differential privacy [19], along with privacy analysis and a bound on the privacy loss of the entire procedure.

This work provides a novel integrated Federated Learning (FL) framework, *FedBNN*, designed to tackle challenges arising from both limited data and heterogeneous computational resources across clients. Additionally, our method offers valuable characterizations of model uncertainties, and is able to operate under strict data privacy constraints, thereby extending the applicability of FL to crucial domains such as healthcare and legal, where these considerations are paramount. To the best of our knowledge, no previous work has jointly addressed all these learning challenges in the FL context. Our promising results significantly broaden the potential of FL for critical real-world applications.

Specifically, our **key contributions** can be summarized as :

- We propose a new approach to personalized federated learning utilizing Bayesian principles for improved robustness and reliability, particularly in contexts where data is scarce. Despite its Bayesian framework, this method is designed to be both computationally and communication efficient.

- A novel collaboration mechanism based on assigning prior distributions over the model parameters via the output space, instead of directly sharing model parameters' distributions, which can be computationally expensive and raise privacy concerns, is proposed to enable clients having different computational resources to train models with varying complexity. This is important because in real-world FL applications, clients often have vastly different capabilities.

- We provide a formal differential privacy guarantee for our method that applies to general settings irrespective of the client's learning algorithm and show that the method is able to learn effectively even under strict privacy guarantees.

- We evaluated our method on several datasets and show that it outperforms the baselines by a significant margin, particularly in heterogeneous data and model settings. This makes FedBNN particularly well-suited for real-world FL applications, which often exhibit high degrees of heterogeneity.

## 2 Related Work

This section provides a brief overview of the most relevant prior work in the fields of federated learning, Bayesian FL, and Differential Privacy in FL.

**Federated Learning** FL was introduced as the FedAvg algorithm in the seminal work in [45]. Since then many different modifications have been proposed that tackle specific challenges, including global FL solutions as well as personalized solutions. FedPD [74], FedSplit [54], and FedDyn [1] proposed methods for finding better fixed-point solutions to the FL optimization problem. [40, 73, 66, 58, 15] show that point-wise aggregate of the local client models does not produce a good global model and propose alternate aggregation mechanisms to achieve collaboration. Personalized FL has been

Table 1: Contrasting our method, FedBNN, against previous works.

| Method | Addressed Challenges | | | |
| --- | --- | --- | --- | --- |
| | Limited Data | Heterogeneous Compute | Uncertainty Quantification | Privacy |
| FedProx | ✗ | ✗ | ✗ | ✓ |
| pFedME | ✓ | ✗ | ✓ | ✗ |
| FOLA | ✓ | ✗ | ✓ | ✗ |
| pFedGP | ✓ | ✗ | ✓ | ✗ |
| pFedBayes | ✓ | ✗ | ✓ | ✗ |
| FedPop | ✓ | ✗ | ✓ | ✗ |
| FedAUX | ✗ | ✓ | ✗ | ✓ |
| **FedBNN** | ✓ | ✓ | ✓ | ✓ |

approached in many ways like meta-learning [20, 8, 31, 33], multi-task learning [59, 38, 60], by clustering the clients [56, 26] and others [17, 37, 72, 57, 67, 43], [16] uses Bayesian view based analysis to obtain a better trade-off between personal and global models. Knowledge distillation for personalized FL settings has also been used previously for training heterogeneous models in non-Bayesian settings [36, 40, 50]. Several other methods have proposed enhancements in FL learning and privacy by using an auxiliary dataset like [15, 55, 36, 52]. Most of these methods rely on the well-established knowledge distillation procedures. But since the transfer of knowledge or information between Bayesian models itself has remained inadequately addressed, these methods are not easily extensible to Bayesian settings. Our method on the other hand, utilizes a novel method that enables collaboration across client specific Bayesian models by transferring knowledge through a prior specification mechanism in the output space, which also enhances the field of Bayesian knowledge distillation.

**Bayesian Federated Learning** Bayesian approaches for federated learning can also be broadly divided as methods using Bayesian inference to obtain a global model and personalized Bayesian learning methods. Amongst the methods that train a global model, some methods just use Bayesian mechanisms for achieving collaboration among non-Bayesian local models, like FedBE [15] which uses Bayesian mechanism to aggregate the locally trained neural networks to obtain a Bayesian ensemble at the server, [9] which suggests using an MCMC based method for obtaining a global model from the local models, and PFNM [73] and FedMA [66] which use a Beta-Bernoullli process to obtain the global models. Other methods that train local Bayesian models at the clients and at the server include FedPA [4] that uses Laplace approximations for an efficient way of computing local and global posteriors, [18] that suggests the use of Bayesian Optimization and Thompson Sampling to obtain the solution to the global optimization problem, recently, [49] did an empirical study on various ways of aggregation mechanisms for local variational Bayesian neural networks and their effects on the solution. These methods that focus on obtaining a global solution are less suited for the statistical heterogeneity present across clients [14], and therefore we focus more on the methods that build personalized Bayesian solutions for clients. Among such methods, pFedGP [3] is a Gaussian Process based estimation method that utilizes Deep Kernel Learning to collaboratively train a single deep neural network with FedAvg and then uses personalized GPs for prediction. FedLoc [70] also uses GP in FL but for regression tasks. pFedBayes [75] uses variational inference locally to optimize a loss at each client that is a combination of the data likelihood term and distance to the prior and iteratively determines the prior from the global posterior distribution. FOLA [41] proposed using Laplace Approximation for posterior inference at both the server side and the client side, PAC-FL [11] and [34, 65, 51] also proposed variants of methods that assume Bayesian models on local clients but for all of them main assumption is that the local model parameters are generated from a shared global distribution thus making them useful only in homogeneous settings. All the methods described above choose priors by assuming a distribution over values for each weight, and thus choosing an appropriate and meaningful prior becomes a challenge [14]. These issues led us to use functional space priors instead which have been explored in limited centralized settings [63, 61, 21] but not in FL. But most importantly, none of these methods are designed or could be easily extended to work with compute heterogeneous settings limiting the applicability of these solutions in several real-world scenarios. Table 1 compares our approach with the most closely related works.

**Differential Privacy in FL** Since decentralized learning does not guarantee that the data will remain private, it is important that a formal rigorous guarantee be given on the data that is leaked by the algorithm. Seminal works in DP propose using a Gaussian noise mechanism by adding Gaussian noise to the intermediate results and achieving a bound on the algorithm by using composition

results [19, 46, 32]. For FL, [24] and [44] independently proposed DP-FedSGD and DP-FedAvg algorithms, which enhance FedAvg by adding Gaussian noise to the local client updates. Several other works focus on analyzing the privacy-utility trade-off in DP in FL setting [25, 27, 5, 64, 39]. Recently, [30] proposed a DP-based solution for personalized FL that works only for linear models. And then [47] improved it for general models and heterogeneous data in FL. These methods, however, mostly focus on privacy guarantees while solving the non-Bayesian FL optimization problem.

## 3 Methodology

In this section, we first go over the problem setting and background, and then present our proposed framework, FedBNN, with details of all the key components.

### 3.1 Background

**Problem Description**    Consider an FL setting with $N$ clients where each client $i$ has local dataset $\mathcal{X}_i$ of size $n_i$ drawn from the local data distribution $\boldsymbol{D}_i$. The goal of a personalized federated learning procedure is to obtain optimal weights for each client's local model, $\mathcal{W}_i^*$, given the entire data, $\mathcal{X} = \bigcup_{j=1}^N \mathcal{X}_j$ through collaboration but without compromising client data privacy. However, the learning procedure faces challenges that are posed due to - *system heterogeneity* and *statistical heterogeneity*. System heterogeneity refers to the variable amount of data and compute resources across clients, meaning, i) the data resources on each client vary widely, i.e., $n_k >> n_l$ for some clients $k$ and $l$, and ii) the compute across clients is non-identical due to which it is not possible to train models of uniform architectures across clients, leading to non-identical weights, i.e., $\mathcal{W}_i \neq \mathcal{W}_j$ for different clients $i$ and $j$. Statistical heterogeneity implies that the data distribution across clients is non-IID.

**Bayesian Learning**    Instead of obtaining the optimal values of the model parameters, $\mathcal{W}_i^*$, Bayesian learning aims to learn posterior distributions (probability distributions over the values) for all the model parameters from the given data - $\mathbb{P}(\mathcal{W}|\mathcal{X})$. Thus, in a personalized Bayesian FL procedure, the modified goal would be to learn distributions for local weights, $\mathbb{P}(\mathcal{W}_i|\mathcal{X})$ from $\mathcal{X} = \bigcup_{j=1}^N \mathcal{X}_j$. However, the exact inference for obtaining the posterior distribution for each of the weight parameter in the network is intractable and several approximations have been studied to obtain approximate distributions. Variational inference [29] is an approximation method that tries to learn parameterized distribution $q(w|\theta)$ from a family of distributions $\mathcal{Q}$, typically of simpler form, by optimizing the parameters $\theta$ such that the new distribution $q(w|\theta^*)$, obtained for the optimal value of $\theta$, is close to the desired posterior distribution $\mathbb{P}(\mathcal{W}|\mathcal{X})$. Precisely, $\theta^*$ is obtained by solving the following optimization problem and its expansion given below -

$$\theta^* = \underset{\theta : q(\mathcal{W}|\theta) \in \mathcal{Q}}{\arg\min} \ \mathrm{KL}[q(\mathcal{W}|\theta)||\mathbb{P}(\mathcal{W}|\mathcal{X})] \tag{1}$$

$$= \underset{\theta : q(\mathcal{W}|\theta) \in \mathcal{Q}}{\arg\min} \ \mathrm{KL}[q(\mathcal{W}|\theta)||p(\mathcal{W};\psi)] - \mathbb{E}_{q(\mathcal{W}|\theta)}[\log\mathbb{P}(\mathcal{X}|\mathcal{W})] \tag{2}$$

and then $q(w|\theta^*)$ is used in place of $\mathbb{P}(\mathcal{W}|\mathcal{X})$. The optimization objective minimizes the distance of $q(\mathcal{W}|\theta)$ to a prior distribution $p(\mathcal{W};\psi)$, used to encode any prior information about the parameters, while also maximizing the likelihood of the observed data $\mathcal{X}$ under $q(\mathcal{W}|\theta)$. A more detailed discussion of Bayesian learning is included in Appendix A. Even though Bayesian approaches are more computationally expensive than their point-estimation counterparts, their superior capabilities for uncertainty quantification and performance in small data settings outweigh the extra compute costs in many critical applications. Moreover, recent innovations like Bayes by Backprop [10] which carefully uses the backpropagated gradients for learning the parameters of posterior distributions, drastically reducing the added computation costs.

### 3.2 FedBNN Methodology

The FedBNN framework works iteratively in two steps - local optimization on the individual clients to obtain local posterior distributions over the model parameters, and a global collaboration step

where the output from each client is appropriately aggregated at the server and broadcast to all the clients for the next rounds of training. These two steps are further described below, and the detailed algorithm and the overview diagram are included in the Appendix B in Algorithm 1 and Figure 2 respectively.

**Local Setting**   Let each client in the network be training a personalized Bayesian NN, which for the client $i$ is denoted by $\Phi_i$ and is parameterised by weights $\mathcal{W}_i$. As commonly used in the literature, we assume that the individual weights of the BNN are Normally distributed and satisfy mean-field decomposition, i.e., $w_{i,\alpha} \sim \mathcal{N}(\mu_{i,\alpha}, \sigma_{i,\alpha}^2)$ for $\alpha \in [1, \ldots, |\mathcal{W}_i|]$ where $\mu_{i,\alpha}$ is the mean of the Gaussian distribution for the parameter $\alpha$ on the $i^{th}$ client and $\sigma_{i,\alpha}^2$ is the variance of the Gaussian distribution for the same parameter. To guarantee that $\sigma_{i,\alpha}$ takes non-negative values for all clients $i$ and all parameters $\alpha$, we use a technique commonly used in inference procedures [10], and replace each $\sigma_{i,\alpha}$ by another parameter $\rho_{i,\alpha}$ during the training, with $\sigma_{i,\alpha} = \log(1 + \exp(\rho_{i,\alpha}))$. The individual weights of the local BNN, $w_{i,\alpha}$, are also assumed to each have a Gaussian prior distribution, $p(w_{i,\alpha}; \psi_{i,\alpha})$, parameterized by $\psi_{i,\alpha} = (\mu_{i,\alpha}^p, \sigma_{i,\alpha}^p)$.

### 3.2.1   Global Collaboration

We attain collaboration amongst clients via an auxiliary dataset called the Alignment Dataset (AD). This is an unlabeled dataset typically small in size, and is used for providing peer supervision to the individual clients by helping clients distill knowledge from other peer clients without explicitly sharing a large number of locally learned parameter weight distributions. The experiments in Figure 5 and Table 3 show the effect of the varying size and distribution of AD in achieving effective collaboration.

In heterogeneous settings, the use of non-identical architecture models ($\mathcal{W}_i \neq \mathcal{W}_j$) means that there is no direct way of aggregating the distributions for prior specification. In fact, even in homogeneous settings, aggregating the weight distributions can be prone to errors due to reasons like insufficient understanding of the weight space, non-alignment of weights across models, etc. Thus, for the purpose of collaboration, we use the function-space of the networks rather than the weight space. Specifically, in each global communication round, the server shares the AD with all the clients. The clients do a forward pass on AD to obtain the local output $\Phi_i(\text{AD})$, where the local output of the $i^{th}$ client is approximated by drawing $m$ sets of weight samples, $\mathcal{W}_i^{(j)} : j \in [1, m]$, from its local posterior distribution $\mathbb{P}(\mathcal{W}_i|\mathcal{X})$ using Monte Carlo sampling and aggregating the outputs under each of these samples $\Phi_i(\text{AD}) = \frac{1}{m} \sum_{j=1}^m \Phi_i(\text{AD}; \mathcal{W}_i^{(j)})$. The obtained output for AD on each client is then sent back to server which forms an aggregated representation, denoted by $\bar{\Phi}(\text{AD})$, obtained via a weighted aggregation of all clients' outputs, i.e., $\bar{\Phi}(\mathbf{X}) = \sum_{j=1}^N w_j \Phi_j(\mathbf{X})$. By default, all weights are considered the same, however the formulation provides flexibility, for example to accommodate situations where the aggregation weights could represent the relative strength of each client in terms of its data or compute resources, i.e., clients with high compute (or data) resources receive more weight as compared to clients with lower amount of resources. The obtained $\bar{\Phi}(\text{AD})$ is then uploaded to all the clients for use in the next round of local training. More details about the Alignment Dataset (AD) along with the explanations and experiments on the size, distribution, availability etc. of the AD are included in the Appendix E.

### 3.2.2   Local Optimization on Clients

**Prior Specification Design**   The Bayesian framework provides a natural way of incorporating supervision in the form of priors. Conventional methods in Bayesian deep learning provide direct priors for model weights as distribution over values. However, the relationship between the values of the model weights/parameters and the outputs is complex and the priors in model's weight-space do not directly capture the desired functional properties. Also, since the number of parameters in a neural network is large, most prior specifications tend to take a simplistic form like an isotropic Gaussian, to make inference feasible. Thus, learning by specifying prior distributions over weights does not always help translate prior knowledge in the learning process. In this work, we consider a way of specifying priors in the functional space by first optimising the Bayesian neural networks over the prior parameters for a fixed number of steps so that the BNN achieves a desired functional output. These intuitive priors help in explicitly instilling the external knowledge during the training of the

neural networks. Let $p(\mathcal{W}_i; \psi)$ represent the prior function over the weights $\mathcal{W}_i$ and is parameterized by $\psi$, with $\psi = \{(\mu_{i,\alpha}^p, \sigma_{i,\alpha}^p), \alpha \in [1, \ldots, |\mathcal{W}_i|]\}$, the prior parameters that determine the prior distributions are learned by solving an optimization problem as below:

$$\psi_i^* = \arg\min_\psi \mathrm{d}(Y, \Phi_i(\mathrm{AD}; \mathcal{W}_i)),$$

where d is a suitable distance function and $Y$ represents the desired output, resulting in optimal priors $p(\mathcal{W}_i; \psi^*)$. Below we provide details of the prior specification for our method.

**Local Optimization** For the local optimization, the individual clients learn $\mathbb{P}(\mathcal{W}_i | \mathcal{X}_i)$ via variational inference. As described above, a variational learning algorithm tries to find optimal parameters $\theta^*$ of a parameterized distribution $q(\mathcal{W}_i | \theta)$ among a family of distributions denoted by $\mathcal{Q}$. In our setting, we set the family of distributions, $\mathcal{Q}$, to be containing distributions of the form $w_{i,\alpha} \sim \mathcal{N}(\mu_{i,\alpha}, \sigma_{i,\alpha}^2)$ for each parameter $w_{i,\alpha}$ for $\alpha \in [1, \ldots, |\mathcal{W}_i|]$. For inference in Bayesian neural networks, we use Bayes by Backprop [10] method to solve the variational inference optimization problem.

At the beginning of each local optimization procedure (in each global communication round a specific client is selected), we use the global information obtained from the server $\bar{\Phi}(\mathrm{AD})$ to intialize the prior for the BNN. Specifically, at the beginning of each local training round, the selected clients first tune their priors to minimize the distance between the local output, $\Phi_i(\mathbf{AD}; \mathcal{W}_i)$ and the aggregated output obtained from the server, $\bar{\Phi}(\mathrm{AD})$. Since the aggregated output represents the collective knowledge of all the clients and may not be *strictly precise* for the local model optimization, we consider this aggregated output as "noisy" and correct it before using for optimization. Specifically, we generate $\Phi_i^{\mathrm{corrected}}$ as a convex combination of the global output and the local output for a tunable parameter $\gamma$. For the $i^{th}$ client,

$$\Phi_i^{\mathrm{corrected}} = \gamma \bar{\Phi}(\mathrm{AD}) + (1 - \gamma)\Phi_i(\mathrm{AD}; \mathcal{W}_i). \tag{3}$$

The prior optimization steps then optimize the distance between $\Phi_i^{\mathrm{corrected}}$ and $\Phi_i(\mathrm{AD}; \mathcal{W}_i)$ to train the prior parameters $\psi$, with the aim of transferring the global knowledge encoded in $\Phi_i^{\mathrm{corrected}}$ to the local model. Precisely,

$$\psi_i^* = \arg\min_\psi \mathrm{d}(\Phi_i^{\mathrm{corrected}}, \Phi_i(\mathrm{AD}; \mathcal{W}_i)). \tag{4}$$

When the outputs $\Phi(\mathrm{X}; \mathcal{W})$ are logits, we use cross-entropy or the negative log-likelihood loss as the distance measure. The optimization involves training the client's personal BNN $\Phi_i$ to only learn the parameters of the prior distribution denoted by $\psi$. This way of initializing the BNN prior enables translating the functional properties, as captured by $\Phi_i(\mathrm{AD}; \mathcal{W}_i)$, to weight-space distributions. The optimal prior parameters are then kept fixed while training the BNN over the local dataset. The local optimization procedure now works to find the best $q(\mathcal{W}_i | \theta)$ fixing the prior distribution through the following optimization problem :

$$\theta_i^* = \arg\min_{\theta: q(\mathcal{W}_i|\theta) \in \mathcal{Q}} \mathrm{KL}[q(\mathcal{W}_i|\theta) || p(\mathcal{W}_i; \psi_i^*)] - \mathbb{E}_{q(\mathcal{W}_i|\theta)}[log\mathbb{P}(\mathcal{X}_i | \mathcal{W}_i)], \tag{5}$$

which is similar to the optimization problem defined in Equation 1 except that now the prior parameters are optimized so that the obtained prior distributions capture the global knowledge and can guide the local learning process to make $q(\mathcal{W}_i | \theta)$ close to the global collective knowledge.

### 3.2.3 Achieving Differential Privacy

In this variation, to control the release of information from the clients, we add a carefully designed Gaussian mechanism wherein we add Gaussian noise to the $\Phi_i(\mathrm{AD})$ that is being shared by each client. Specifically, each client $i$ uploads $\Phi_i(\mathrm{AD})_{\mathrm{DP}} = \Phi_i(\mathrm{AD}) + \mathcal{N}(0, \sigma_g^2)$ to the server and then the server aggregates $\Phi_i(\mathrm{AD})_{\mathrm{DP}}$ across clients to obtain and broadcast $\bar{\Phi}(\mathrm{AD})_{\mathrm{DP}}$ which is used by the clients in their next round of local optimization. The variance of the noise depends on the required privacy guarantee.

# 4 Privacy Analysis

Though our algorithm is inherently quite private as it refrains from explicitly sharing model weights, we can also provide a formal Differential Privacy based guarantee. Our analysis in this section focuses on providing record-level DP guarantee over the entire dataset $\mathcal{X}$. This analysis quantifies the level of privacy achieved towards any third party and an honest-but-curious server. In this section we directly present the key result of our analysis. Due to the lack of space, additional definitions, results and the proof for the theorem are mentioned in Appendix C.

**Theorem 4.1** (Privacy Budget). *The proposed algorithm is $(\epsilon, \delta)$-differentially private, if the total privacy budget per global communication round per query is set to*

$$\rho = \frac{\epsilon^2}{4EK log \frac{1}{\delta}}$$

*for $E$ number of global communication rounds and $K$ number of queries to the algorithm per round.*

The parameter $\rho$ is related to the Gaussian noise by $\rho = \frac{\Delta^2}{2\sigma^2}$. The detailed proof is included in Appendix C. Our analysis does not assume any specifics of how each client is trained and is therefore applicable in more general settings. Note that we present a pessimistic analysis by providing a worst-case analytical bound, wherein we assume that a change in single data point may entirely change the output of the algorithm, and also since the public dataset remains common throughout the rounds, the actual privacy loss due to querying on the public dataset does not typically add up linearly. Yet the above analysis shows that we have several knobs to control to achieve the desired privacy-utility trade off - balancing the number of global communication rounds with local epochs, reducing the number of queries, and the standard noise scale. By appropriately tuning these controls we are able to achieve good performance with a *single digit* $\epsilon$ ($\approx 9.98$) privacy guarantee and $\delta = 10^{-4}$.

# 5 Experiments

In this section, we present an experimental evaluation of our method and compare it with different baselines under diverse homogeneous and heterogeneous client settings. Specifically, we experiment with three types of heterogeneity - i) heterogeneity in data resources (amount of data), ii) heterogeneity in compute resources, and iii) statistical heterogeneity (non-IID data distribution across clients). We also discuss the change in performance of our method when the degree and type of heterogeneity changes. Due to the space constraint, additional experiments on varying the size and distribution of the AD, privacy-utility trade-off and model calibration are included in the Appendix E, G and D respectively.

## 5.1 Experimental Details

**Datasets** We choose three different datasets commonly used in prior federated learning works from the popular FL benchmark, LEAF [13] including MNIST, CIFAR-10 and CIFAR-100. MNIST contains 10 different classes corresponding to the 10 digits with 50,000 $28 \times 28$ black and white train images and 10,000 images for validation. CIFAR-10 and CIFAR-100 contain 50,000 train and 10,000 test-colored images for 10 classes and 100 classes respectively. The choice of these datasets is primarily motivated by their use in the baseline methods.

**Simulation Details** We simulate three different types of heterogeneous settings - corresponding to heterogeneity in compute resources, data resources and the statistical data distribution. Before starting the training process, we create $N$ different clients with different compute resources by randomly selecting a fraction of clients that represent clients with smaller compute. Since these clients do not have large memory and compute capacity, we assume that these clients train smaller-size BNNs as opposed to the other high-capacity clients that train larger VGG-based models. In particular, the small BNNs were constructed to have either 2 or 3 convolution layers, each followed by a ReLU and 2 fully-connected layers at the end, and a VGG9-based architecture was used for larger BNNs. The number of parameters in smaller networks is around 50K and that in larger networks is around 3M. Since the baselines only operate with identical model architectures across clients, we use the larger VGG9-based models on the baselines for a fair comparison. We include the results of our

method in both homogeneous compute settings (similar to baselines) as well as in heterogeneous compute settings wherein we assume that 30% of the total clients have smaller compute and are training smaller-sized models.

Next, we also vary the data resources across clients and test the methods under 3 different data settings - small, medium and full. The small setting corresponds to each client having only 50 training data instances per class, for the medium and full settings each client has 100 data instances and all available data instances per class respectively for training. We simulate statistical heterogeneity by creating non-IID data partitions across clients. We work in a rather strict non-IID setting by assuming clients have access to data of disjoint classes. For each client a fraction of instance classes is sampled and then instances corresponding to the selected classes are divided amongst the specific clients. For the included experiments, we set number of clients $N = 20$ and divide the instances on clients such that each client has access to only 5 of the 10 classes for MNIST and CIFAR-10, and 20 out of 100 classes for CIFAR-100.

Table 2: Test accuracy comparsion with baselines in non-IID settings.

| Method | MNIST | | | CIFAR10 | | | CIFAR100 | | |
|---|---|---|---|---|---|---|---|---|---|
| | (small) | (medium) | (full) | (small) | (medium) | (full) | (small) | (medium) | (full) |
| (Non-Bayesian) | | | | | | | | | |
| Local Training | $88.7 \pm 1.2$ | $90.1 \pm 1.0$ | $91.9 \pm 1.1$ | $53.9 \pm 2.1$ | $59.5 \pm 1.8$ | $70.8 \pm 1.4$ | $28.8 \pm 1.8$ | $32.7 \pm 1.9$ | $43.5 \pm 1.6$ |
| FedAvg | $88.2 \pm 0.5$ | $90.15 \pm 1.2$ | $92.23 \pm 1.0$ | $43.14 \pm 1.2$ | $56.27 \pm 1.8$ | $78.17 \pm 1.2$ | $27.3 \pm 1.9$ | $32.81 \pm 1.6$ | $36.3 \pm 0.2$ |
| FedProx | $86.9 \pm 0.8$ | $89.91 \pm 0.7$ | $93.1 \pm 0.4$ | $44.27 \pm 1.2$ | $58.93 \pm 0.9$ | $79.19 \pm 0.6$ | $28.6 \pm 2.7$ | $34.31 \pm 1.4$ | $37.8 \pm 0.9$ |
| FedAUX | $90.1 \pm 1.6$ | $92.8 \pm 1.34$ | $94.4 \pm 1.21$ | $60.01 \pm 1.96$ | $68.6 \pm 0.73$ | $77.0 \pm 0.84$ | $37.05 \pm 1.3$ | $43.5 \pm 1.7$ | $45.2 \pm 0.88$ |
| pFedME | $91.95 \pm 2.1$ | $93.39 \pm 1.2$ | $95.62 \pm 0.5$ | $48.46 \pm 1.5$ | $64.57 \pm 2.1$ | $75.11 \pm 1.2$ | $32.4 \pm 2.2$ | $36.3 \pm 2.0$ | $41.8 \pm 1.7$ |
| non-Bayesian KD | $89.1 \pm 0.4$ | $92.5 \pm 0.2$ | $93.2 \pm 0.3$ | $33.9 \pm 1.3$ | $53.2 \pm 1.5$ | $69.8 \pm 1.0$ | $26.1 \pm 2.0$ | $35.2 \pm 1.2$ | $42.7 \pm 0.8$ |
| (Bayesian with Homogeneous Architectures) | | | | | | | | | |
| pFedGP | $86.15 \pm 1.3$ | $90.59 \pm 1.7$ | $94.92 \pm 0.3$ | $45.62 \pm 2.2$ | $56.24 \pm 1.8$ | $72.89 \pm 0.7$ | $47.06 \pm 1.3$ | $53.1 \pm 1.2$ | $54.54 \pm 0.2$ |
| pFedBayes | $94.0 \pm 0.2$ | $94.6 \pm 0.1$ | $95.5 \pm 0.3$ | $58.7 \pm 1.1$ | $64.6 \pm 0.8$ | $78.3 \pm 0.5$ | $39.51 \pm 1.8$ | $41.43 \pm 0.4$ | $47.67 \pm 1.1$ |
| FOLA | $91.74 \pm 1.0$ | $92.87 \pm 0.8$ | $95.12 \pm 0.6$ | $43.29 \pm 0.9$ | $45.94 \pm 0.7$ | $67.98 \pm 0.5$ | $33.42 \pm 1.3$ | $48.8 \pm 2.1$ | $43.2 \pm 1.6$ |
| Ours (Homo) | $\mathbf{94.9 \pm 1.0}$ | $\mathbf{95.72 \pm 0.8}$ | $\mathbf{96.21 \pm 0.3}$ | $\mathbf{70.6 \pm 1.1}$ | $\mathbf{72.3 \pm 0.6}$ | $\mathbf{79.7 \pm 0.3}$ | $\mathbf{49.65 \pm 1.4}$ | $\mathbf{55.4 \pm 0.8}$ | $\mathbf{57.3 \pm 0.8}$ |
| Ours (Hetero) | $93.1 \pm 1.1$ | $94.4 \pm 0.2$ | $95.9 \pm 0.2$ | $\mathbf{68.17 \pm 2.0}$ | $\mathbf{71.73 \pm 1.3}$ | $78.7 \pm 0.7$ | $\mathbf{47.5 \pm 1.4}$ | $49.10 \pm 1.1$ | $\mathbf{51.1 \pm 0.7}$ |
| Ours (Hetero-DP) | $89.82 \pm 2.3$ | $90.21 \pm 1.6$ | $91.43 \pm 1.4$ | $54.9 \pm 1.91$ | $61.83 \pm 1.4$ | $74.3 \pm 1.6$ | $43.7 \pm 2.3$ | $44.5 \pm 1.7$ | $47.0 \pm 1.5$ |
| (DP-Baseline) | | | | | | | | | |
| DP-FedAvg | $80.1 \pm 1.7$ | $85.2 \pm 1.8$ | $86.2 \pm 1.7$ | $35.17 \pm 0.8$ | $50.22 \pm 1.1$ | $74.6 \pm 1.2$ | $26.5 \pm 0.3$ | $30.7 \pm 1.4$ | $32.4 \pm 0.6$ |

**Training parameters and Evaluation** We run all the algorithms for 200 global communication rounds and report the accuracy on the test dataset at the end of the $200^{th}$ round. The number of local epochs is set to 20 and the size of AD is kept as 2000. Each client is allowed to train its personal model for a fixed number of epochs, which is kept to 50 in experiments, before entering the collaboration phase. The hyper-parameters of the training procedure are tuned on a set-aside validation set. At the beginning of each global communication round, for optimizing the prior parameters at each client according to Equation 4, we use an Adam optimizer with learning rate=0.0001 and run the prior optimization procedure for 100 steps. Then with the optimized prior we train the local BNN using Bayes-by-Backprop, with Adam optimizer, learning rate = 0.001 and batch size = 128. The noise effect $\gamma$ is selected after fine-tuning and kept to be 0.7. For these experiments, the aggregation weight $w_j$ for each client $j$ used to compute $\bar{\Phi}(\mathbf{X})$ is set to $1/N$, and the AD is obtained by using a separated subset of the dataset in consideration. All the models are trained on a 4 GPU machine with GeForce RTX 3090 GPUs and 24GB per GPU memory. For evaluation, we report the classification accuracy obtained by running the trained models on test datasets from the MNIST, CIFAR10 and CIFAR100 datasets.

**Baselines** We compare our method against the standard non-Bayesian FL algorithms and Bayesian-FL methods that build personalized models for clients. We also show results of differentially private FedAvg algorithm under similar privacy guarantee to provide perspective on the privacy. Apart from local training where all clients train independent models locally without collaboration, the non-Bayesian FL baselines include - i) FedAvg, ii) FedProx, iii) pFedME (which uses personalized models on each client using Monreau envelopes in loss). We also compare our method to other

baselines that use an auxiliary dataset for collaboration in non-Bayesian FL, namely i) FedAUX, which uses federated distillation for achieving collaboration in FL, we do not use FedMD [36] as a baseline since it requires a labelled auxiliary dataset. Further, we create a baseline corresponding to the non-Bayesian version of our method that works with knowledge distillation and call it non-Bayesian KD. The Bayesian FL baselines include - i) pFedGP, a Gaussian process based approach that trains common deep kernels across clients and personal tree-based GPs for classification, ii) pFedBayes, which uses a variational inference-based approach for personalized FL by training personal models which are close to the aggregated global models, iii) FOLA, bayesian method using Gaussian product for model aggregation. And lastly, the DP baseline includes - i) DP-FedAvg, the FedAvg algorithm with gradient clipping and noise addition to the gradient at each client. The size of the AD is changed to 1000, number of local epochs to 40 and global communication rounds to 100 for DP-based experiments. For all the experiments, the hyper-parameters were obtained by tuning on a held-out validation dataset. We used our own implementation of the pFedBayes algorithm since the source code was not publicly available but we could not compare against FedPop due to the lack of some implementation details and publicly unavailable code.

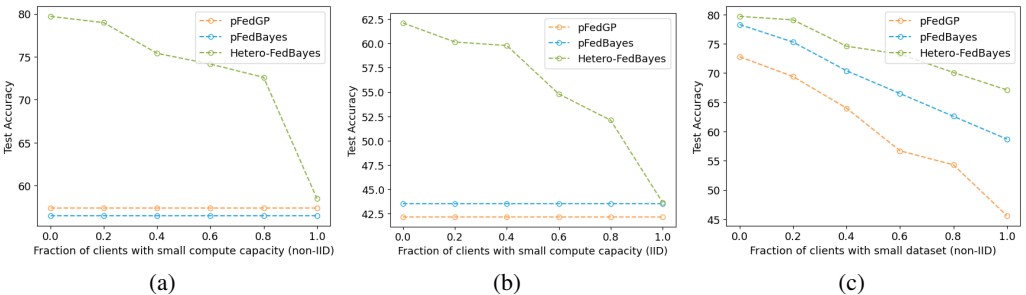

Figure 1: Performance comparison of our method with baselines under different types and varying degree of heterogeneity for CIFAR-10 dataset with 20 clients. Figure (a) is for heterogeneity in compute capacity across clients under non-IID data setting, figure (b) for compute heterogeneity under IID setting, and figure (c) for heterogeneity in data resources. When a fraction of clients in the setting have low computing resources, the baselines being homogeneous can only train smaller models on all the clients as shown by constant performance. The results show that our method is more tolerant to both model heterogeneity and data heterogeneity across clients.

## 5.2 Results

The performance of our method and the baselines under the non-IID data setting are reported in Table 2. Under the non-IID setting, we report the results corresponding to different dataset sizes on each client. To recall, in the small, medium, and full settings, each client has access to 50, 100, and all training data points per class respectively. We observe that our method with homogeneous architectures across clients outperforms all other baselines. Moreover, when we consider the performance of our method under a heterogeneous setting by considering 30% of the total clients to be small capacity, it is evident that our method is better than the higher capacity homogeneous baselines for more complex tasks like in CIFAR-10 and CIFAR-100. On average, our method achieves about $6\%$ performance improvement over the baselines in the small and medium data settings. Figure 1 compares the performance of our method with the highest-performing baselines under model, data and statistical types of heterogeneity. Since our method can work with heterogeneous clients, we see that just by the proposed collaboration and having higher capacity clients in the FL ecosystem, the lower capacity clients are able to gain about $10\%$ increase in their performance. Also, the performance degradation of our method with a change in the number of clients with limited data resources is more graceful as compared to the baselines. In an additional experiment intended to compare the performance of the baseline methods with additional data, we trained the priors for baseline methods' encoders using the unlabeled data, AD, before starting their own prescribed FL procedure. We observed that the performance of the baseline methods does not change on doing this because the FL procedure that they incorporate forgets all the prior existing local knowledge at the client side. A similar result was also reported in [55]. The superior performance of our method could be attributed to the innovative and effective collaboration achieved by first distilling peer knowledge in the form of the aggregated output on the

AD, and then ensuring that this knowledge is successfully transferred to each client by specifying priors in the functional-space of the client model. Furthermore, the parameter in Equation 3 allows the clients the flexibility to choose the amount of global knowledge that needs to be incorporated, providing flexibility on the degree of personalization.

# 6  Discussion

This paper introduced a novel method for personalized Bayesian learning in heterogeneous FL settings and demonstrated that it is able to outperform existing approaches under different types of heterogeneous situations, while also providing a privacy guarantee and calibrated responses. The experiments show that the method is particularly useful for clients with lower data and lower compute resources as they can benefit the most by the presence of other, more powerful clients in the ecosystem. While our method assumes the availability of a small, unlabelled auxiliary dataset at the server, it is typically a very mild requirement as such data can often be obtained from several open sources on the web. In many cross-silo and cross-device applications, the server often possesses its dataset alongside private data from clients. For example, hospitals with access to patient records may combine this data with private patient data collected from individual devices such as wearables or sensors for FL, source code generation applications might leverage open-source code along with private code repositories from developers, etc. [6, 7] also mention use-cases where such data is available in real-world. Recent advances in generative AI have made creation of synthetic data for training a much easier task. The privacy analysis on the method provides an intuitive and a rigorous guarantee with various tunable knobs that can be adjusted to achieve the desired privacy-utility trade-off. And while the application explored in the proposed work consists of image related tasks, both the proposed framework and the privacy analysis are generic and independent of specific training algorithms, therefore resulting in its wide applicability in various applications across data modalities. Also, while Bayesian methods are inherently more computationally expensive as they have to maintain distributions rather than point estimates, this extra work is invaluable in many applications where uncertainty quantification is important, for example to help engineers account for uncertainties in material properties, loading conditions, and manufacturing processes, leading to safer and more reliable designs [53]. The recent use of transformer based Bayesian methods [76, 42, 62] in varied applications indicate that the proposed framework can be also applied to settings where much larger neural networks are required. One limitation that originates from the Bayesian nature, and is common to all applications of Bayesian learning, is that the exact inference of the posterior distributions is infeasible and therefore variational approximation has been used for inference of the posterior distributions.

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

# Supplement for "A Bayesian Approach for Personalized Federated Learning in Heterogeneous Settings"

In this supplementary material, we first go over the preliminaries of Bayesian learning methods, followed by the pseudo-code of the algorithm used for training our framework. Then, we provide definitions and results used in the privacy analysis of the method along with the proof of our privacy budget theorem. We show model calibration metrics and present results demonstrating our method is well-calibrated. We also discuss the details about the alignment dataset, AD, its affect on the performance, include additional experimental results and discuss the communication and computation cost of the procedure.

## A  Bayesian Learning

Consider a learning setting where we are trying to train a neural network on a dataset $\mathcal{X}$. The aim of this setting is thus to obtain the set of weights, denoted by $\mathcal{W}$, for the corresponding neural network that best fits the data. We could also view a neural network as a model that outputs $\mathbb{P}(y|x, \mathcal{W})$ which is the distribution of the label $y$ for a given data point $x$ under the weights $\mathcal{W}$, for classification this would be the output of the softmax function. Now, the weights of the network can be learnt by Maximum Likelihood Estimation (MLE) for a given set of datapoints $\mathcal{X} = (x_i, y_i)_{i=1}^n$ by solving the following optimization problem.

$$\mathcal{W}^{MLE} = \arg\max_{\mathcal{W}} \sum_i \log P(y_i|x_i, \mathcal{W})$$

This optimization could be solved by gradient descent based methods and obtains a point estimate of the weight vector, denoted by $\mathcal{W}^{MLE}$.

The Bayesian learning methods, on the other hand, obtain a posterior distribution on the weights given the training data, $\mathbb{P}(\mathcal{W}|\mathcal{X})$, which as opposed to the point estimates denotes the joint distribution of all the weight parameters of the network over the set of values they are likely to take under the observed data and the prior information encoded in the prior distribution. The predictions for any new data point, $x$, are then obtained by taking expectation of the prediction under the posterior distribution, $y = \mathbb{E}_{w \sim \mathbb{P}(\mathcal{W}|\mathcal{X})}[\mathbb{P}(y|x, w)]$. Exact inference of the posterior distribution, however, is intractable for neural networks. Variational inference is a traditional approximation method used to obtain an approximation of the posterior weight distribution, and it has also been shown to work for neural networks [29]. Specifically, variational inference tries to learn a simpler parameterized distribution $q(w|\theta)$ from a family of distributions $\mathcal{Q}$ by optimizing the parameters $\theta$ such that the new distribution $q(w|\theta^*)$ obtained for the optimal value of $\theta$ is close to the true posterior distribution $\mathbb{P}(\mathcal{W}|\mathcal{X})$. Precisely, the optimization problem looks like

$$\theta^* = \arg\min_{\theta: q(\mathcal{W}|\theta) \in \mathcal{Q}} \text{KL}[q(\mathcal{W}|\theta)||\mathbb{P}(\mathcal{W}|\mathcal{X})] \tag{6}$$

$$= \arg\min_{\theta: q(\mathcal{W}|\theta) \in \mathcal{Q}} \int q(\mathcal{W}|\theta) \log \frac{q(\mathcal{W}|\theta)}{\mathbb{P}(\mathcal{W})\mathbb{P}(\mathcal{X}|\mathcal{W})} \tag{7}$$

$$= \arg\min_{\theta: q(\mathcal{W}|\theta) \in \mathcal{Q}} \text{KL}[q(\mathcal{W}|\theta)||p(\mathcal{W}; \psi)] - \mathbb{E}_{q(\mathcal{W}|\theta)}[\log \mathbb{P}(\mathcal{X}|\mathcal{W})] \tag{8}$$

where $p(\mathcal{W}; \psi)$ signifies the prior distribution over weights $\mathcal{W}$ parameterized by $\psi$. The prior distribution is typically used to encode any previously available information about the weights of the network. The above given objective is the same objective as in Equation 5 that is used for local training in our method.

## B  Algorithm

The pseudo-code of the algorithm used in the FedBNN method is included in the Algorithm 1. The Algorithm 1 works in the setting when there is a server connected to $N$ clients with each client $i$ having local dataset $\mathcal{X}_i$ of size $n_i$ drawn from the local data distribution $\boldsymbol{D}_i$, and the server has an auxilliary unlabelled dataset called AD. The output of the algorithm is the set of personalized models $\Phi_i$ parameterized by $\mathcal{W}_i$ for each client $i$. All $\mathcal{W}_i$'s, instead of being point estimates, are determined by a posterior distribution $\mathbb{P}(\mathcal{W}_i|.)$ which is learnt from the data via variational inference. As mentioned in the Section 3.2, the learning procedure first optimizes the prior parameters by minimizing Equation 4 and then learns the posterior parameters keeping the prior fixed by minimizing Equation 5.

## C  Privacy Analysis

Some known results on differential privacy that are used to determine the privacy loss of our algorithm are given in this section and then the proof of the Theorem 4.1 is presented.

**Algorithm 1** FedBNN Algorithm

---

**Input:** number of clients $N$, number of global communication rounds $E$, number of local epochs $e$, weight vector $[w_1, w_2, \ldots w_N]$, noise parameter $\gamma$

**Output:** Personalized BNNs $\{\Phi_i | i \in [1, N]\}$, parameterized by $\mathcal{W}_i \sim \mathbb{P}(\mathcal{W}_i | \mathcal{X})$

**Server Side -**

$\mathbf{X} = \text{AD}$

**for** $t = 1$ **to** $E$ **do**

    Select a subset of clients $\mathcal{N}_t$

    **for** each selected client $i \in \mathcal{N}_t$ **do**

        $\Phi_i(\mathbf{X}) = \textbf{LocalTraining}(t, \bar{\Phi}(\mathbf{X})^{(t-1)}, \mathbf{X})$

    **end for**

    $\bar{\Phi}(\mathbf{X})^{(t)} = \sum_{j=1}^{\mathcal{N}_t} w_j \Phi_j(\mathbf{X})$

**end for**

Return $\Phi_1(E), \Phi_2(E) \ldots \Phi_N(E)$

---

$\textbf{LocalTraining}(t, \bar{\Phi}(\mathbf{X})^{(t-1)}, \mathbf{X})$

Run inference on $\mathbf{X}$ to obtain $\Phi_i(\mathbf{X})$

Generate $\Phi_i^{\text{corrected}}(\mathbf{X}) = \gamma \bar{\Phi}(\mathbf{X})^{(t-1)} + (1 - \gamma)\Phi_i(\mathbf{X})$

**for** each prior epoch **do**

    Minimize CrossEntropy($\Phi_i^{\text{corrected}}(\mathbf{X}), \Phi_i(\mathbf{X})$) to obtain prior parameters $\psi$ of the BNN $\Phi_i$

**end for**

**for** each local epoch **do**

    Minimize $\text{KL}[q(\mathcal{W}_i | \theta) || p(\mathcal{W}_i; \psi^*)] - \mathbb{E}_{q(\mathcal{W}_i | \theta)}[log\mathbb{P}(\mathcal{X}_i | \mathcal{W}_i)]$ over $\{\theta : q(\mathcal{W}_i | \theta) \in \mathcal{Q}\}$ to obtain $\theta^*$

**end for**

$\mathbb{P}(\mathcal{W}_i | \mathcal{X}) \approx q(\mathcal{W}_i | \theta^*)$

Obtain $m$ Monte-carlo samples $\mathcal{W}_i^{(j)} : j \in [1, m]$ from $\mathbb{P}(\mathcal{W}_i | \mathcal{X})$

Compute $\Phi_i(\mathbf{X}) = \dfrac{1}{m} \sum_{j=1}^{m} \Phi_i(\mathbf{X}; \mathcal{W}_i^{(j)})$

Return $\Phi_i(\mathbf{X})$

---

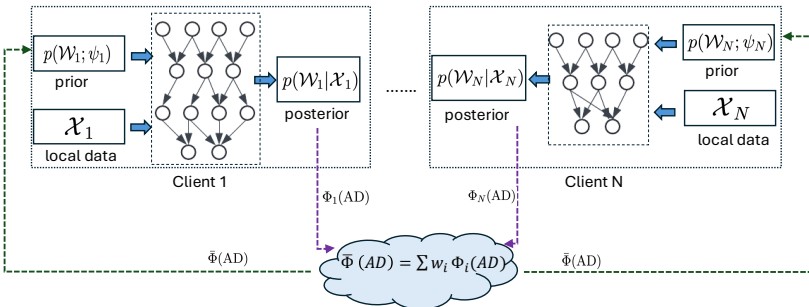

Figure 2: A schematic overview of our method: Local BNN on each client obtain a posterior distribution over local parameters using the prior distribution and local data. These local models generate outputs on the AD using their respective posterior distributions and share these outputs with the server. The server aggregates these outputs and distributes the aggregated output on the AD back to all clients, guiding the prior distribution on each client. These updated prior distributions then further guide the learning of the posterior distributions.

**Definition C.1** (($\epsilon, \delta$)- Differential Privacy). A randomized algorithm $\mathcal{M}$ is ($\epsilon, \delta$)-DP if for any two neighboring datasets $D$ and $D'$ that differ in at most one data point, the output of the algorithm $\mathcal{M}$ on $D$ and $D'$ is bounded as

$$\mathbb{P}[\mathcal{M}(D) \in S] \leq e^\epsilon \mathbb{P}[\mathcal{M}(D') \in S] + \delta, \quad \forall S \subseteq \text{Range}(\mathcal{M}).$$

A generalization of differential privacy is known as concentrated differential privacy(CDP). And an alternative form of concentrated differential privacy called zero-concentrated differential privacy(zCDP) was proposed to enable tighter privacy analysis [12]. We will also use the zCDP notion of privacy for our analysis. The relationship between standard DP and zCDP is shown below.

**Proposition C.2** (($\epsilon, \delta$)-DP and $\rho$-zCDP). *For a randomized algorithm $\mathcal{M}$ to satisfy ($\epsilon, \delta$)-DP, it is sufficient for it to satisfy $\frac{\epsilon^2}{4log\frac{1}{\delta}}$-zCDP. And a randomized algorithm $\mathcal{M}$ that satisfies $\rho$-zCDP, also satisfies ($\epsilon', \delta$)-DP where $\epsilon' = \rho + \sqrt{4\rho log\frac{1}{\delta}}$.*

As opposed to the notion of DP, the zCDP definition provides tighter bounds for the total privacy loss under compositions, allowing better choice of the noise parameters. The privacy loss under the serial composition and parallel composition incurred under the definition of zCDP was proved by [71] and is recalled below.

**Proposition C.3** (Sequential Composition). *Consider two randomized mechanisms, $\mathcal{M}_1$ and $\mathcal{M}_2$, if $\mathcal{M}_1$ is $\rho_1$-zCDP and $\mathcal{M}_2$ is $\rho_2$-zCDP, then their sequesntial composition given by $(\mathcal{M}_1(), \mathcal{M}_2())$ is $(\rho_1 + \rho_2)$-zCDP.*

**Proposition C.4** (Parallel Composition). *Let a mechanism $\mathcal{M}$ consists of a sequence of $k$ adaptive mechanisms, $(\mathcal{M}_1, \mathcal{M}_2, \ldots \mathcal{M}_k)$ working on a randomized partition of the $D = (D_1, D_2, \ldots D_k)$, such that each mechanism $\mathcal{M}_i$ is $\rho_i$-zCDP and $\mathcal{M}_t : \prod_{j=1}^{t-1} \mathcal{O}_j \times D_t \rightarrow O_t$, then $\mathcal{M}(D) = (\mathcal{M}_1(D_1), \mathcal{M}_2(D_1), \ldots \mathcal{M}_k(D_k))$ is $\max_i \rho_i$-zCDP.*

After computing the total privacy loss by an algorithm using the tools described above, we can determine the variance of the noise parameter $\sigma$ for a set privacy budget. The relationship of the noise variance to privacy has been shown in prior works by [19, 71] and is given below.

**Definition C.5** ($L_2$ Sensitivity). For any two neighboring datasets, $D$ and $D'$ that differ in at most one data point, $L_2$ sensitivity of a mechanism $\mathcal{M}$ is given by maximum change in the $L_2$ norm of the output of $\mathcal{M}$ on these two neighboring datasets

$$\Delta_2(\mathcal{M}) = \sup_{D, D'} ||\mathcal{M}(D) - \mathcal{M}(D')||_2.$$

**Proposition C.6** (Gaussian Mechanism). *Consider a mechanism $\mathcal{M}$ with $L_2$ sensitivity $\Delta$, if on a query $q$, the output of $\mathcal{M}$ is given as $\mathcal{M}(x) = q(x) + \mathcal{N}(0, \sigma^2)$, then $\mathcal{M}$ is $\frac{\Delta^2}{2\sigma^2}$-zCDP.*

Equipped with the above definitions and results, we now re-state the bound on the privacy loss of our algorithm and provide a proof below.

**Theorem C.7** (Privacy Budget). *The proposed algorithm is ($\epsilon, \delta$)-differentially private, if the total privacy budget per global communication round per query is set to*

$$\rho = \frac{\epsilon^2}{4EK log\frac{1}{\delta}}$$

*for $E$ number of global communication rounds and $K$ number of queries to the algorithm per round.*

*Proof.* After using Gaussian mechanism on each client and adding noise to each coordinate of $\Phi_i(AD)$, the local mechanism at each client becomes $\rho$-zCDP for $\rho = \frac{\Delta^2}{2\sigma^2}$. Since each client outputs the logit representation for each input, i.e., the normalized output of the clients, $\Delta^2 \leq 2$. The sensitivity, denoted as $\Delta$, is defined in Definition C.4 in the paper which defines $L_2$-sensitivity as the maximum change in the $L_2$ norm of the algorithm's output between two neighboring datasets differing in at most one data point. Let $D$ and $D'$ be two neighboring datasets that differ in one data point present at the $i^{th}$ row (without loss of generality), and let $\Phi(D(i, :))$ be the $n_c$ (number of classes) dimensional output probabilities from the model $\Phi$ for the $i^{th}$ row datapoint in $D$ and $\Phi(D'(i, :))$ be the output probabilities for the $i^{th}$ row datapoint in $D'$. The $L_2$ sensitivity of $\Phi$ is -

$$\Delta(\Phi) = ||\Phi(D) - \Phi(D')||_2$$

Since all other data-points between $D$ and $D'$ are identical, the $L_2$ sensitivity of $\Phi$ becomes -

$$\Delta(\Phi) = ||\Phi(D(i, :)) - \Phi(D'(i, :))||_2$$

Now, $\Phi(D(i, :))$ and $\Phi(D'(i, :))$ are both probability distributions, therefore it can be seen that the squared $L_2$ norm of their difference is bounded by 2, i.e., $\Delta(\Phi)^2 \leq 2$ (the maximum occurs when $\Phi(D(i, :))_k = 1$ and $\Phi(D(i, :))_l = 1$ for two separate indices $k \neq l$).

Now, suppose in each global communication round we make $K$ queries to each client, then by sequential composition C.3, we get $EK\rho$, for $E$ number of global communication rounds. By parallel composition C.4, the total privacy loss for all $N$ clients is the maximum of the loss on each client and therefore remains $EK\rho$.

Relating it to the $(\epsilon, \delta)$-DP from C.2, we get $\rho = \dfrac{\epsilon^2}{4EK\log\frac{1}{\delta}}$ for any $\delta > 0$. $\qquad\square$

# D Uncertainty Quantification and Calibration

Model calibration is a way to determine how well the model's predicted probability estimates the model's true likelihood for that prediction. Well-calibrated models are much more important when the model decision is used in critical applications like health, legal etc. because in those cases managing risks and taking calculated actions require a confidence guarantee as well. Visual tools such as reliability diagrams are often used to determine if a model is calibrated or not. In a reliability diagram, model's accuracy on the samples is plotted against the confidence. A perfectly calibrated model results in an identity relationship. Other numerical metrics that could be used to measure model calibration include Expected Calibration Error (ECE) and Maximum Calibration Error (MCE). ECE measures the expected difference between model confidence and model accuracy whereas MCE measures the maximum deviation between the accuracy and the confidence. The definitions and empirical formulas used for calculating ECE and MCE are as given below.

$$\text{ECE} = \mathbb{E}_{\hat{P}}[\mathbb{P}(\hat{Y} = Y | \hat{P} = p) - p],$$

$$\text{MCE} = \max_{p \in [0,1]} |\mathbb{P}(\hat{Y} = Y | \hat{P} = p) - p|.$$

Empirically,

$$\text{ECE} = \sum_{i=1}^{M} \frac{|B_i|}{n} |\text{accuracy}(B_i) - \text{confidence}(B_i)|,$$

$$\text{MCE} = \max_{i \in [1,M]} |\text{accuracy}(B_i) - \text{confidence}(B_i)|,$$

where $B_i$ is a bin with set of indices whose prediction confidence according to the model falls into the range $\left(\frac{i-1}{M}, \frac{i}{M}\right)$. Figure 3 shows the reliability diagram along with the ECE and MCE scores for our method measured on MNIST and CIFAR-10 dataset in the non-IID data setting.

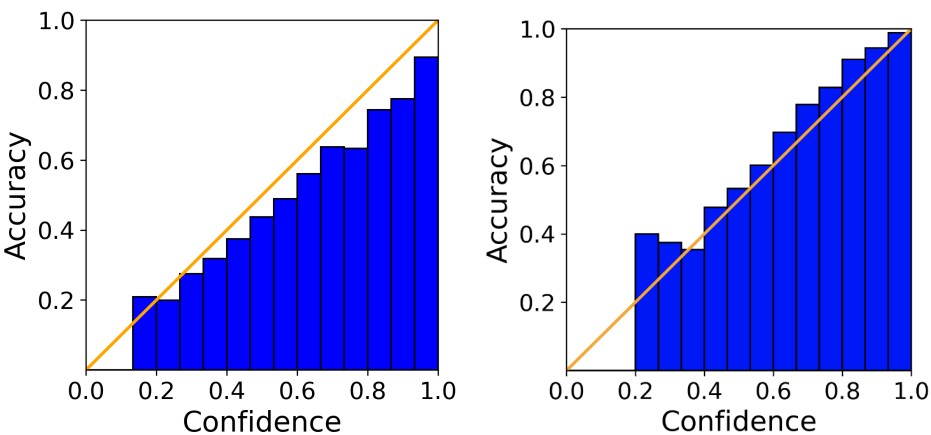

(a) **Dataset**: CIFAR-10, **ECE**: 0.070, **MCE**: 0.134   (b) **Dataset**: MNIST, **ECE**: 0.032, **MCE**: 0.156

Figure 3: Reliability diagrams and scores showing model calibration. Figure (a) is for the results corresponding to the CIFAR-10 dataset and Figure (b) for MNIST dataset.

For the next analysis, we consider an approach similar to analysis on uncertainty quantification done in exemplar works in this area [35, 48]. One of the key requirement of reliable estimates and uncertainty quantification is ensuring high confidence in correct predictions and low confidence in incorrect predictions. To assess whether the proposed method meets this criterion, we train our method using standard MNIST train dataset, and test it on the MNIST test dataset as well as on an out-of-distribution dataset composed of NotMNIST10 (featuring images of alphabets instead of digits). Then, we compute the entropy of the predictive distribution (distribution over output class probabilities) for each dataset and visualize this entropy in the provided Figure **??**. In the first row corresponding to the in-distribution dataset, both our Bayesian model and the non-Bayesian model exhibit

low entropy, as expected. However, for the out-of-distribution test dataset, while the non-Bayesian method demonstrates low entropy, our method yields high entropy. This observation implies that the non-Bayesian method tends to be overly confident in its predictions on unknown classes, which could pose significant risks in practical scenarios, especially in critical applications.

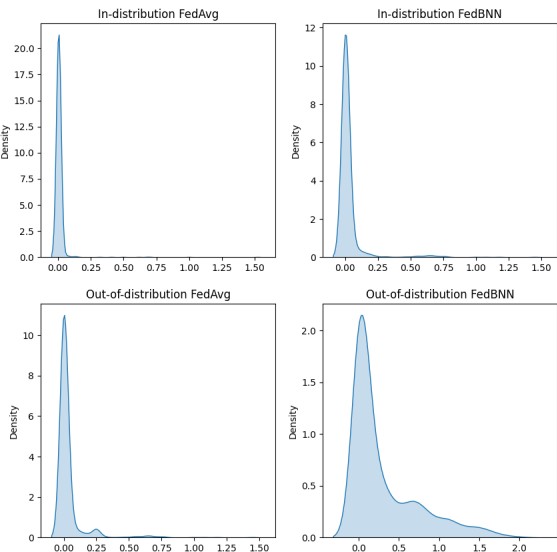

Figure 4: Distribution of the entropy of class-probability distributions across different clients demonstrating the confidence of methods in predicting on in-distribution vs out-of-distribution data.

# E   Alignment Dataset (AD)

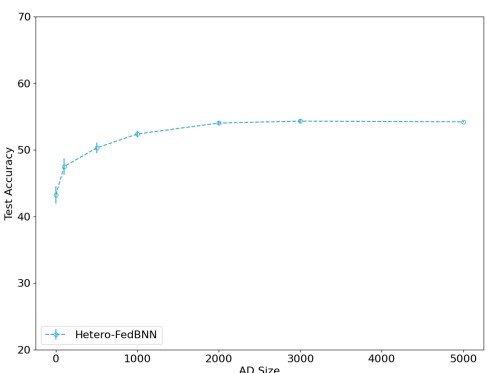

Figure 5: Ablation study comparing the affect of AD size on the performance. The included results are for CIFAR-10 dataset in the small data setting with non-IID partitions and heterogeneous clients.

In FedBNN, the alignment dataset (AD) is used to achieve collaboration across clients. Since the only assumption on AD is for it to be of the same domain as the target application, there is no practical constraint on obtaining the AD in real-world settings. In many cases it could be obtained from web, for example images from common datasets in Huggingface, texts from Wikipedia, Reddit etc. Furthermore, the server having its own dataset in addition to the private data on the clients is common in various cross-silo and cross-device applications. For instance, hospitals with access to patient records may combine this data with private patient data collected from individual devices such as wearables or sensors for federated learning, source code generation applications might leverage open-source code along with private code repositories from developers to enhance generative models, self-driving car companies may collect their own data and utilize it alongside private data collected from customers' vehicles, and many more. The use of AD is not different from how several other methods use an additional dataset for augmentation, and several other methods have used a labelled or an unlabelled auxiliary

Table 3: Effect of varying distribution of AD on the clients' performance for the non-IID seeting with CIFAR-10 dataset and 20 clients where each client has data for the 5 different classes.

| Architecture Setting | Local Training | CIFAR10(10) | CIFAR10(8) | CIFAR10(5) | CIFAR10(2) | SVHN |
|---|---|---|---|---|---|---|
| Homogeneous Architectures | $64.3 \pm 0.36$ | $72.7 \pm 0.15$ | $69.7 \pm 0.28$ | $68.8 \pm 0.97$ | $67.2 \pm 1.5$ | $70.1 \pm 0.18$ |
| Heterogeneous Architectures | $61.2 \pm 0.17$ | $71.6 \pm 0.93$ | $68.4 \pm 0.80$ | $68.8 \pm 1.4$ | $68.1 \pm 1.9$ | $69.3 \pm 0.8$ |

dataset to improve the performance and privacy of the FL algorithms [52, 36, 55]. The effect of size of AD on the performance of models is demonstrated in Figure 5 for CIFAR-10 dataset in the small data and non-IID setting. In that figure, we observe that when the size of AD is small the performance of the model is low but as the size of AD increases the performance increases up to a point and becomes constant afterwards. The number of data points in AD that are required to achieve good improvement in the model performance is small and practical.

We also vary the distribution of the AD being used and test the final performance of the models and report it in Table 3. We run these experiments on 20 clients for CIFAR-10 dataset where each client had access to only 5 of the 10 classes and each client belonged to the medium data setting. For the first experiment, we use a held-out dataset from the CIFAR-10 data as AD but vary the composition of the dataset by changing the distribution of the classes present in the AD, for example, CIFAR10(10) is composed of all 10 classes present in the CIFAR-10 dataset but CIFAR10(2) is composed of only 2 out of the 10 classes present in the AD and likewise. We also test the performance of our method when a significantly different dataset SVHN consisting of the colored house number images is used. Table 3 suggests that the performance of the method even with different datasets as AD *always* improves and that the gain between local training and the proposed procedure is better highlighted in the heterogeneous architecture settings, since there local client capacities and model architectures differ significantly and clients are able to utilize the peer knowledge to learn better models locally. We observed that even for different and dissimilar data distributions in AD, it is possible to obtain a value for the parameter $\gamma$ such that the final performance of the local client model with collaboration is better than the model independently trained locally on the client. The best results for CIFAR-10 classification are seen when AD is composed of a held-out set from all 10 classes of CIFAR-10 denoted as CIFAR10(10) which is as expected. Then, as the composition of AD is changed from 10 classes to random 8, 5 and 2 classes of CIFAR-10 (denoted as CIFAR10(8), CIFAR10(5) and CIFAR10(2) respectively) the performance keeps on decreasing. We see that the performance of the same task with SVHN as AD is only strictly better than CIFAR10(2) as AD. We believe that the SVHN dataset as AD works better than the CIFAR10(2) because it provides more variability in the data distribution. A similar observation was also recorded in FedAUX [55] which also uses additional unlabelled data for knowledge distillation, which noted that the out-of-domain unlabelled data for distillation can perform even better. Moreover, the parameter $\gamma$ controls the amount of global knowledge to be incorporated on each client and with appropriately set $\gamma$, the AD also helps achieve regularization for the local client model such that the local models do not overfit to the relatively smaller local datasets and generalize better.

## F   Communication and Computation Efficiency

**Communication Cost**   In FedBNN, each global communication round requires that the server sends the alignment dataset to all the clients and the clients upload the outputs of their respective models on the common dataset AD. Since AD is a publicly available dataset, AD could be transmitted to the clients by specifying the source and the indices, and does not really needs to be communicated across the channel. The client output on AD, on the other hand, depends on the number of instances in AD, let's call it $K$, therefore, the total communication cost in each round of our method is $O(K)$. As shown in Figure 5, having $K = 2000$ gives a good performance. The communication cost between the clients and the server, thus, is also invariant of the number of model parameters which tends to run in millions. This allows our method to be much more communication efficient as compared to the conventional FL algorithms and other Bayesian FL methods that transmit model parameters or parameter distributions in each communication round, making it practically more useful.

**Computation Cost**   Similarly, the computation cost of a FL procedure involves the costs incurred in local training at the individual clients and the cost of aggregation at the server, both of which are discussed below.

- **Server-side computation cost** The server side computation cost arises from the need to aggregate knowledge obtained from individual clients. In the state-of-the-art bayesian FL algorithms, the server aggregates posterior distributions for each weight parameter in the neural network obtained from various clients. The number of such weight parameters typically run in millions. In our method we do not aggregate the parameter distributions but achieve collaboration by aggregating the client outputs on the AD (with size  2000), thus the server side computation cost in our method is many orders

Table 4: Performance comparison as a function of the privacy guarantee.

| Privacy ($\epsilon$) per round | Test Accuracy |
|---|---|
| $\approx 1$ | 75.5 % |
| $\approx 0.1$ | 71.3 % |
| $\approx 0.01$ | 68.6 % |
| $\approx 0.001$ | 62.2 % |
| $\approx 0.0001$ | 59.6 % |

Table 5: Test accuracy comparison with more number of clients (500) in the setting.

| Method | Test Accuracy |
|---|---|
| pFedGP | $53.2 \pm 0.4$ |
| pFedBayes | $52.9 \pm 0.8$ |
| Ours(Homo) | $56.1 \pm 0.3$ |
| Ours(Hetero) | $54.7 \pm 1.0$ |

of magnitude lower than the conventional methods and does not depend on the number of model parameters. This makes our method much more efficient and scalable than existing federated bayesian solutions.

- **Client-side computation cost** The client-side computation cost is mostly determined by the cost of training a Bayesian Neural Network at the client side, which in turn depends on the type of inference procedure used for obtaining the posterior distribution over weights for each parameter of the neural network. In the proposed work, the method used for inference is Bayes by Backprop which uses the gradient computations similar to backpropagation(BP) algorithm to obtain the posterior distributions where the posterior distributions are characterized by the mean and standard deviation. A re-parameterization trick is used to compute the mean and std of the distributions from the back-propagated gradients. Thus the cost for obtaining the posterior distributions is similar to the cost of backpropagation. Moreover, since the method only uses gradient updates, the optimizations used for SGD like asynchronous SGD etc. could be readily used for obtaining the posteriors. An unrelated but similar algorithm in [28] does probabilistic backpropagation to train BNNs and shows that the average run time of probabilistic BP is not higher than that of BP.

To summarize, the communication cost and the server-side computation cost of the proposed method is orders of magnitude lower than that of the other Bayesian baseline methods. On the other hand, the client-side computation cost is determined by the inference procedure used to obtain the posterior distributions and for which Bayes by Backprop provides an efficient mechanism. Several works in the recent past have discussed the use of related Bayesian inference based methods for training uncertainty-aware transformers [76, 62, 42] proving that Bayesian methods are not limited to use in simpler models. And therefore, our framework can also be extended to apply in settings where much larger neural networks are required.

## G  Additional Experiments

**Privacy vs Performance**  Since the amount of noise required to be added to the client's outputs via the Gaussian Mechanism is directly proportional to the guaranteed privacy, we test the affect of the privacy guarantee on the performance of the proposed framework by comparing the performance of the method with varying $\epsilon$ and $\delta = 10^{-4}$. The results are reported in Table 4. We observe that, as expected, when we reduce the amount of privacy loss in each iteration by adding more noise to the clients' outputs going to the server, the performance of the method drops. However the drop in performance in all the cases is not drastic as the clients can tune the level of personalization or global knowledge required by appropriately setting the parameter $\gamma$ in Equation 3.

**More clients**  To test the performance of the proposed method when a large number of clients are involved in the setup, we did additional experiments with 500 clients and non-IID setting with 5 classes per client in the medium data setting on the CIFAR-10 dataset where in each communication round only $10\%$ of the clients are selected for participation and $\gamma = 0.7$. The obtained results at the end of $200^{th}$ communication round are reported in Table 5. We observe that the homogeneous version of our method is better than the baselines by a significant margin and the heterogeneous version is slightly better than the baselines.

