# OpenReview forum: "A Bayesian Approach for Personalized Federated Learning in Heterogeneous Settings"
_NeurIPS.cc/2024/Conference — NeurIPS 2024 poster_

### Official Review · Reviewer_PtiT · 2024-07-12

**Soundness:** 2
**Presentation:** 2
**Contribution:** 2
**Rating:** 4
**Confidence:** 3

**Summary:**

The authors designed a federated learning algorithm that leverages a Bayesian network to enhance the model's robustness, and uses data distillation methods to reduce overhead and adapt to scenarios with different client model structures. The authors theoretically discussed the differential privacy characteristics of the method and demonstrated its effectiveness through experiments.

**Strengths:**

1. The paper formally discusses the characteristics of differential privacy, ensuring privacy protection.
2. Experiments about distillation dataset sizes are quite insightful.

**Weaknesses:**

1. The authors stated that the paper has two contributions: 1) to address the issue of small data volumes per client by using a Bayesian network; 2) to solve the overhead problem brought by the Bayesian network and the model heterogeneity issue by using a distillation method. However, these two methods seem relatively independent and both have their own rich related works. It appears the authors simply combined these two works directly, resembling an A+B work, which is less compelling.
2. The authors used a distillation scheme, but the distillation scheme requires the server to maintain public datasets. However, in many scenarios, this is very difficult to achieve, which is a major criticism of similar distillation methods.
3. The experiments are insufficient. As a paper that mainly measures performance through experimental results, the experimental part of the main text is only 2.5 pages (of which 1.5 pages are explanations of the experimental setup). Although the authors added some additional experiments in the appendix, it is still far from sufficient, and the overall experiment was completed under only one setting. It is recommended that the authors at least consider adding the following experiments: 1) Since the paper involves data heterogeneity, different data heterogeneity scenarios (such as Dirichlet non-IID) should be tested; 2) Expand to more common neural network structures instead of using the authors' self-constructed simple CNNs. Even if the client's computing power is insufficient, lightweight networks such as MobileNet should be considered.
4. From the perspective of paper presentation, this paper has considerable room for improvement, such as: 1) The paper does not provide a framework diagram of the method, making it very difficult to understand the author's method; 2) The writing style is not concise and clear enough, such as the second point in the contributions; 3) The margin between formulas and text, the width of tables, are not well adjusted.

**Questions:**

1. Please introduce the difficulties and improvements in integrating the Bayesian network into the data distillation-based federated learning framework.
2. What is the performance of this method under other data heterogeneity or model structures?

**Limitations:**

The authors discussed possible social impact and application and also touched on the potential limitations of the method.

---

> ### Author Rebuttal · Authors · 2024-08-06
>
> We appreciate reviewer's insightful comments and suggestions. We are encouraged by reviewer's positive remarks about the formal privacy guarantee and the experiments on the alignment dataset.
>
> We address reviewer's specific concerns below -
>
> -- *Clarification on the contribution*
>
> We agree with the reviewer that the bayesian framework and the knowledge distillation have been independently integrated into FL by previous works. However, our contributions are particularly significant for the following reasons -
>
> 1.  *Knowledge distillation in Bayesian learning itself is non-trivial*. While knowledge distillation is a well-established method, the transfer of knowledge between Bayesian models has remained inadequately addressed. Our work introduces a novel method that facilitates collaboration across client-specific Bayesian models by transferring knowledge through a prior specification mechanism in the output space. This contribution significantly advances the field of Bayesian knowledge distillation, in both centralized as well as federated learning settings.
> 2. *Bayesian learning across heterogeneous clients in FL is novel.* The designed collaboration mechanism, is pivotal in enabling efficient collaboration across models of heterogeneous architectures (encompassing varying size and shape), significantly advancing the FL and Bayesian-FL procedures for heterogeneous settings. By means of such a procedure, the small compute capacity clients are able to gain about 10 increase in the performance through collaboration with large compute capacity clients (as shown in Table 2 and Figure 1 in the paper). This is in contrast to the conventional Bayesian-FL procedures that work with homogeneous models and would either leave out the small compute clients out of the FL process or reduce the model capacity on the large compute clients leading to under-performance.
> 3. *Incorporating a privacy-preserving Bayesian FL mechanism is novel*. As the reviewer correctly points out our method also incorporates privacy along with Bayesian learning in heterogeneous FL settings, making it more usable for a diverse set of critical applications.
>
> -- *On model structures in Experiments*
>
> We would like to kindly clarify to the reviewer that we did not only use "self-constructed simple CNNs" for the experiments, as mentioned in the review. In fact, we used a combination of popular VGG-based models and CNNs to demonstrate the heterogeneity in our setting. This approach highlights the stability and performance of our method under such drastic heterogeneity. Also, under the homogeneous client setting in the experiments (detailed in subsection 5.1 of the paper) all the clients train VGG based models, which aligns with the reviewer's comment on using "more common neural network structures".
>
> -- *On different data heterogeneity in Experiments*
>
> Thanks to the reviewer for this suggestion, we include more experiments on varying degree of heterogeneity below. Given the limited time for author response, we compare the performance of our method under varying degrees of heterogeneity against key Bayesian and the most competitive Bayesian baselines identified from previous experiments. To evaluate performance under different degrees of heterogeneity, we utilize the Dirichlet distribution. Specifically, for each class in the dataset, we distribute the data among 20 clients according to a Dirichlet distribution with parameter $\alpha$, smaller values of  $\alpha$ create more heterogeneous distributions, while larger values create more homogeneous distributions. We conduct this experiment on the CIFAR-10 dataset with a 20-client setting, and report the results in the table below. Note that when  $\alpha$ is low, the number of classes per client is less than 10, leading to an easier task and higher accuracy for all methods compared to the setting when  $\alpha$ is high, and all 10 classes are present on each client, resulting in a more complex task. We have also added these new results to the revised version of the paper.
>
> | Method                 | $\alpha$ = 0.1 | $\alpha$ = 1   | $\alpha$ = 10  |
> | ---------------------- | -------------- | -------------- | -------------- |
> | Local Training         | 70.1 $\pm$ 1.5 | 54.5 $\pm$ 1.8 | 47.3 $\pm$ 0.7 |
> | FedAvg (non-Bayesian)  | 66.2 $\pm$ 1.4 | 53.7 $\pm$ 1.2 | 49.0 $\pm$ 0.5 |
> | FedProx (non-Bayesian) | 66.9 $\pm$ 1.1 | 56.8 $\pm$ 0.9 | 50.1 $\pm$ 0.9 |
> | pFedGP (Bayesian)      | 65.7 $\pm$ 1.2 | 58.3 $\pm$ 1.9 | 51.8 $\pm$ 1.1 |
> | pFedBayes (Bayesian)   | 69.3 $\pm$ 1.4 | 60.1 $\pm$ 1.7 | 52.2 $\pm$ 1.2 |
> | FedBNN (Ours)          | 72.3 $\pm$ 1.9 | 62.5 $\pm$ 1.4 | 54.3 $\pm$ 0.7 |
>
>
>
> -- *Presentation Issues*
>
> We sincerely thank the reviewer for these valuable suggestions. We have included a new figure providing an overview of our method in the revised version of the paper. For the reviewer's reference, the figure and its caption are available at this link: https://i.imgur.com/ZBC2sMu.png.
>
> We also apologize for the discrepancies between the margin and text in certain sections. These adjustments were made to fit the content within the 9-page limit. With the additional page allowed for the camera-ready version, these issues will be easily resolved.

---

> > ### Comment · Reviewer_PtiT · 2024-08-11
> >
> > Overall, I believe the authors' rebuttal has provided some valuable new information. My updated comments are as follows:
> >
> > **Experiments**
> > I appreciate the additional experimental data, which enhances the paper. However, the scope remains somewhat limited, with one dataset and a small number of clients. While I acknowledge the authors' efforts during the rebuttal period, I believe the experiments could be further strengthened.
> >
> > **Model Structure and Presentation Issues**
> > Thank you for the clarifications. I apologize for any earlier misunderstandings. These explanations are helpful, and I look forward to seeing them incorporated into future revisions of the paper.
> >
> > **Novelty**
> > I feel that the authors did not directly address my concerns regarding the novelty (e.g., concern 1). While I agree that some of the innovative aspects highlighted in the rebuttal are indeed non-trivial and novel to some extent, as I mentioned in my first weakness point, there are two issues: 1) Each point, when considered individually, lacks sufficient insight or difficulty, and the approach seems relatively straightforward (in my opinion); 2) When taken together, these innovative points do not seem to be well connected, making the overall contribution less compelling.
> >
> > In summary, while some concerns have been addressed, others remain unresolved, leading me to adjust my score from 3 to 4. I hope this feedback is helpful as the authors continue through the review process.

---

> ### Author Response · Authors · 2024-08-12
> **Response to Official Comment by Reviewer PtiT**
>
> Dear Reviewer PtiT,
>
> We thank you for your detailed response to our rebuttal and sincerely appreciate your thoughtful comments. We further hope to effectively address your concerns  on novelty and additional experiments through the points we make below.
>
> -- *Additional Experiments*
>
> We are glad to hear that you believe the additional experimental results enhance the paper and we thank you again for suggesting these new experiments. Although the limited time during the rebuttal period prevented us from completing the additionally suggested experiments on more datasets and settings, we want to assure you that we have already established the necessary experimental setup which will allow us to easily incorporate results on more datasets and clients in the next revision, and we are fully committed to doing so. We appreciate your understanding and look forward to sharing these updated findings in the final submission.
>
>
>
> -- *Novelty*
>
> We apologize if it seemed we did not fully address your concerns about the novelty of our work. We would like to take this opportunity to provide further clarification.
>
> The key motivation of this work is to enhance the applicability of FL across diverse real-world settings by addressing several interconnected and prevalent challenges in federated environments. These challenges include - i) the issue of small data volume per client, ii) compute heterogeneity across clients, (both of which are known common limitations in federated learning environments), iii) the need for uncertainty quantification to measure and manage uncertainty in predictions, which is crucial for critical applications like in healthcare and legal domains, and iv) the need for privacy preservation which is essential for ensuring the confidentiality of sensitive information and is also important in critical domains. To tackle these issues, we have developed an integrated end-to-end framework, in which each client performs local training using Bayesian learning with its own tailored BNN architecture (and that could be different across clients), and the collaboration is achieved by the novel mechanism of knowledge distillation through prior specification. We also emphasize on the fact that the conventional Bayesian learning methods, whether centralized or federated, often focus on priors in the weight space, which makes them restricted for utilization in FL settings where the clients have heterogeneous data and compute resources, and is also communicationally intensive. Thus, our framework is designed to effectively address these issues as well—issues that would arise from merely applying conventional Bayesian learning methods in FL—through a novel collaboration mechanism.
>
> We recognize that some components of our approach may seem individually straightforward when viewed in isolation. However, it’s crucial to understand that our approach is not just a collection of techniques but a carefully designed system that addresses multiple challenges in FL. It's appropriate formulation into a cohesive and effective framework, without introducing additional complications, for personalized federated learning represents a significant advancement in the field. We hope this further explanation clarifies the novelty and impact of our contributions, and we appreciate your thoughtful consideration of our response.
>
> In essence, the key contributions of our work are not just in - *"1) to address the issue of small data volumes per client by using a Bayesian network; 2) to solve the overhead problem brought by the Bayesian network and the model heterogeneity issue by using a distillation method."*. We thank the reviewer for highlighting the importance of this discussion in the paper, and we will use the additional space available in the final manuscript to include this discussion.

---

> > ### Author Response · Authors · 2024-08-14
> > **Kindly Request for Reviewer's Feedback**
> >
> > Dear Reviewer PtiT,
> >
> > We sincerely appreciate the time you have taken to provide feedback on our work, which has helped us greatly improve its clarity, among other attributes. This is a gentle reminder that the Author-Reviewer Discussion period ends at 11:59 pm AoE on August 13. We are happy to answer any further questions you may have before then, but we will be unable to respond after that time.
> >
> > If you agree that our responses to your reviews have addressed the questions you listed, we kindly ask that you consider whether raising your score would more accurately reflect your updated evaluation of our paper. Thank you again for your time and thoughtful comments!
> >
> > Sincerely,
> >
> > Authors

---

### Official Review · Reviewer_4fxY · 2024-07-13

**Soundness:** 3
**Presentation:** 3
**Contribution:** 3
**Rating:** 5
**Confidence:** 3

**Summary:**

The authors propose FedBNN, a novel personalized federated learning (FL) framework that leverages Bayesian principles to address challenges posed by heterogeneous data and computational resources among clients. FedBNN uses Bayesian neural networks to enable robust local training on small datasets by quantifying uncertainties. A novel collaboration mechanism involves sharing priors in the functional space of the networks, rather than directly sharing model parameters, to accommodate clients with varying computational capacities. The approach also includes a differentially private version with formal privacy guarantees. Experiments on standard FL datasets demonstrate that FedBNN outperforms existing methods, particularly in heterogeneous settings and under strict privacy constraints. The main contributions include improved robustness and efficiency in personalized FL, a novel collaboration method using functional space priors, and a formal differential privacy guarantee applicable to general settings.

**Strengths:**

The paper presents a comprehensive evaluation of their FedBNN framework across multiple dimensions. The authors demonstrate its effectiveness on three major datasets (MNIST, CIFAR-10, CIFAR-100) under various heterogeneous settings, comparing against both Bayesian and non-Bayesian baselines. They provide detailed ablation studies to justify their design choices, including the use of functional space priors and the auxiliary dataset for collaboration. The experiments thoroughly explore the method's performance under different types of heterogeneity (data resources, compute resources, and statistical distribution), showcasing its robustness and adaptability. The authors also present a formal privacy analysis, demonstrating the framework's applicability in privacy-sensitive scenarios. Overall, the extensive experimental evaluation, coupled with the theoretical foundations, provides strong evidence for the effectiveness and significance of their proposed approach in addressing real-world challenges in federated learning.

**Weaknesses:**

While the authors present an innovative Bayesian approach to personalized federated learning (FL), several improvements are needed. The novelty is somewhat diminished by existing methods like FedPop [1]; clearer differentiation and more comprehensive comparisons are necessary. The experimental evaluation is robust but lacks diversity in dataset types and real-world applications; incorporating more varied and complex datasets would better demonstrate generalizability. Scalability concerns are not adequately addressed, particularly regarding the computational overhead of Bayesian neural networks. The privacy analysis should delve deeper into the trade-offs between noise levels and model accuracy. More thorough ablation studies are needed to isolate the impact of individual framework components, such as functional space priors versus traditional weight space priors.
[1] Kotelevskii, N., Vono, M., Durmus, A., & Moulines, E. (2022). Fedpop: A bayesian approach for personalised federated learning. Advances in Neural Information Processing Systems, 35, 8687-8701.

**Questions:**

See Weaknesses above.

---

> ### Author Rebuttal · Authors · 2024-08-06
>
> We appreciate the reviewer’s detailed feedback. We are encouraged to see that the reviewer acknowledges the strength of our work, stating that "*Overall, the extensive experimental evaluation, coupled with the theoretical foundations, provides strong evidence for the effectiveness and significance of their proposed approach in addressing real-world challenges in federated learning*".
>
> We address reviewer's key concerns below -
>
> -- *Differentitaion with FedPop*
>
> While a capability wise contrast of our method with the existing methods including FedPop is included in the Table 2 in Appendix, here we provide a detailed comparison of our method with the reference suggested by the reviewer, named FedPop.
>
> 1. *FedPop can only work in homogeneous FL settings as opposed to our method that can work in heterogeneous settings too*. The FedPop method in [1] assumes a hierarchical statistical model for personalised FL across clients. The hierarchical model constitutes of an unknown population parameter also referred to as prior (denoted by β) from which the individual local parameters for each client are sampled (denoted by z1,z2,..) where the variance of the population parameter determines the heterogeneity of the client specific parameters. Since the client's local parameters are sampled from the same distribution (also called prior), they have identical shape and form and therefore these methods work only in settings when the clients architectures are homogeneous. This is also evident in the aggregation mechanisms used in the algorithms  where they perform element wise aggregation over the parameters. Since the client's local parameters are sampled from the same distribution (the prior), they have an identical shape and form. As a result, these methods are effective only in settings where the clients' architectures are homogeneous. This is also evident in the aggregation mechanisms used in the algorithms, which perform element-wise aggregation over the parameters. However, the problem setting we address involves Bayesian learning in a heterogeneous FL setting, where clients have different computational and data resources. In such a scenario, it is not feasible to utilize FedPop.
> 2. *Our method is much more communication efficient*. Since the FedPop method involves transmitting model parameters between clients and the server in each communication round, its communication cost is proportional to the number of model parameters, which can run into millions. In contrast, our method only transmits the outputs on the alignment dataset between clients and the server in each communication round. This significantly reduces the communication cost, as it involves outputs for approximately 5000 data points, which is much smaller in size.
> 3. *Incorporating a privacy-preserving Bayesian FL mechanism is novel* Our work presents a privacy-preserving method for Bayesian FL along with a bound on privacy loss, no privacy analysis or privacy-preserving algorithm is included in FedPop.
>
> -- *Privacy Analysis*
>
> We clarify to the author that the trade-off between model accuracy and privacy guarantee is included in Table 4 in the Appendix.  Since, there is a direct relationship between the noise parameter and the privacy budget as established in Theorem 4.1, the results in Table 4 are also applicable for the trade-off between model accuracy and noise parameter.
>
> -- *Functional space priors versus traditional weight space priors*
>
> We appreciate the reviewer's suggestion for more thorough ablation studies to isolate the impact of individual framework components, such as functional space priors versus traditional weight space priors.  Knowledge transfer between Bayesian models through priors in functional space is one of the key contributions of this work. These functional space priors are essential to our method, enabling effective knowledge transfer and collaboration across client-specific models in heterogeneous federated learning settings.
>
> To identify the effect of functional space priors versus weight space priors, we can compare our method using functional space priors against a baseline with traditional weight space priors. We direct the reviewer to the results in Table 1, which compares the pFedBayes method (which uses the weight soace priors) and our method in the homogeneous setting. Additionally, the results in Figures 1(a) and 1(b) can be used to compare the impact of functional space and weight space priors across varying degrees of heterogeneity among clients. These results could be used to demonstrate the impact of functional space priors in achieving superior performance.

---

> > ### Author Response · Authors · 2024-08-14
> > **Kindly Request for Reviewer's Feedback**
> >
> > Dear Reviewer 4fxY,
> >
> > We sincerely appreciate the time you have taken to provide feedback on our work, which has helped us greatly improve its clarity, among other attributes. This is a gentle reminder that the Author-Reviewer Discussion period ends at 11:59 pm AoE on August 13. We are happy to answer any further questions you may have before then, but we will be unable to respond after that time.
> >
> > If you agree that our responses to your reviews have addressed the questions you listed, we kindly ask that you consider whether raising your score would more accurately reflect your updated evaluation of our paper. Thank you again for your time and thoughtful comments!
> >
> > Sincerely,
> >
> > Authors

---

### Official Review · Reviewer_yjRB · 2024-07-14

**Soundness:** 2
**Presentation:** 3
**Contribution:** 3
**Rating:** 6
**Confidence:** 3

**Summary:**

This work integrates Bayesian neural networks, knowledge distillation, and differential privacy into the federated learning framework. The resulting framework can maintain privacy during training and provide uncertainty estimates for its predictions. Empirical results demonstrate that the proposed method outperforms previous works in the given experimental setting.

**Strengths:**

1. This work is well-written. The authors have clearly explained the motivation, methodology, and experimental settings.
2. This work addresses data and system heterogeneity and privacy, touching on all the major challenges of federated learning.

**Weaknesses:**

1. Bayesian frameworks and knowledge distillation have been independently integrated into federated learning and studied by many works. Although this work combines them with differential privacy effectively and shows good results, it has limited novelty as an academic research paper.
2. There are limited experimental results. The main results are presented in Table 1, where the experimental setting considers 20 clients and label distribution shift. Given the complexity of practical situations, the authors could consider other experimental settings to demonstrate the generality of the method, such as different types of data heterogeneity, varying numbers of clients, and different numbers of data points across clients. I note that in the appendix, the authors also provide results for 500 clients, but there are only a few numbers. A more comprehensive study similar to Table 1 would be preferable.

**Questions:**

Since only features of the public dataset are transferred, this framework should be communication efficient. The authors could consider plotting accuracy versus transmitted bits to emphasize the strength of the proposed method.

---

> ### Author Rebuttal · Authors · 2024-08-06
>
> We appreciate the reviewer’s valuable feedback. We are pleased to see that the reviewer finds the work well-written, noting that "*The authors have clearly explained the motivation, methodology, and experimental settings*" and recognizing that "*This work addresses data and system heterogeneity and privacy, touching on all the major challenges of federated learning".
>
> We address reviewer's key concerns below -
>
> -- *"Bayesian frameworks and knowledge distillation have been independently integrated into federated learning and studied by many works. "*
>
> We agree with the reviewer that the bayesian framework and the knowledge distillation have been independently integrated into FL by previous works. However, our contributions are particularly significant for the following reasons -
>
> 1.  *Knowledge distillation in Bayesian learning itself is non-trivial*. While knowledge distillation is a well-established method, the transfer of knowledge between Bayesian models has remained inadequately addressed. Our work introduces a novel method that facilitates collaboration across client-specific Bayesian models by transferring knowledge through a prior specification mechanism in the output space. This contribution significantly advances the field of Bayesian knowledge distillation, in both centralized as well as federated learning settings.
> 2. *Bayesian learning across heterogeneous clients in FL is novel.* The designed collaboration mechanism, is pivotal in enabling efficient collaboration across models of heterogeneous architectures (encompassing varying size and shape), significantly advancing the FL and Bayesian-FL procedures for heterogeneous settings. By means of such a procedure, the small compute capacity clients are able to gain about 10 increase in the performance through collaboration with large compute capacity clients (as shown in Table 2 and Figure 1 in the paper). This is in contrast to the conventional Bayesian-FL procedures that work with homogeneous models and would either leave out the small compute clients out of the FL process or reduce the model capacity on the large compute clients leading to under-performance.
> 3. *Incorporating a privacy-preserving Bayesian FL mechanism is novel*. As the reviewer correctly points out our method also incorporates privacy along with Bayesian learning in heterogeneous FL settings, making it more usable for a diverse set of critical applications.
>
> -- *"Experimental Results"*
>
> We would like to clarify to the reviewer that apart from the results in Table 1 and Appendix, we also present results across many different settings in Figure 1, where performance is compared across varying degree of heterogeneity in the non-IID and IID-seeting. For example, like the reviewer said, performance with different number of data points across clients is depicted in Figure 1(c) of the paper, and the varying compute heterogeneity is depicted in Figure 1(a) and  Figure 1(b).
>
> Thanks to the reviewer for this suggestion, we include more experiments on varying degree of heterogeneity below. Given the limited time for author response, we compare the performance of our method under varying degrees of heterogeneity against key Bayesian and the most competitive Bayesian baselines identified from previous experiments. To evaluate performance under different degrees of heterogeneity, we utilize the Dirichlet distribution. Specifically, for each class in the dataset, we distribute the data among 20 clients according to a Dirichlet distribution with parameter $\alpha$, smaller values of  $\alpha$ create more heterogeneous distributions, while larger values create more homogeneous distributions. We conduct this experiment on the CIFAR-10 dataset with a 20-client setting, and report the results in the table below. Note that when  $\alpha$ is low, the number of classes per client is less than 10, leading to an easier task and higher accuracy for all methods compared to the setting when  $\alpha$ is high, and all 10 classes are present on each client, resulting in a more complex task. We have also added these new results to the revised version of the paper.
>
> | Method                 | $\alpha$ = 0.1 | $\alpha$ = 1   | $\alpha$ = 10  |
> | ---------------------- | -------------- | -------------- | -------------- |
> | Local Training         | 70.1 $\pm$ 1.5 | 54.5 $\pm$ 1.8 | 47.3 $\pm$ 0.7 |
> | FedAvg (non-Bayesian)  | 66.2 $\pm$ 1.4 | 53.7 $\pm$ 1.2 | 49.0 $\pm$ 0.5 |
> | FedProx (non-Bayesian) | 66.9 $\pm$ 1.1 | 56.8 $\pm$ 0.9 | 50.1 $\pm$ 0.9 |
> | pFedGP (Bayesian)      | 65.7 $\pm$ 1.2 | 58.3 $\pm$ 1.9 | 51.8 $\pm$ 1.1 |
> | pFedBayes (Bayesian)   | 69.3 $\pm$ 1.4 | 60.1 $\pm$ 1.7 | 52.2 $\pm$ 1.2 |
> | FedBNN (Ours)          | 72.3 $\pm$ 1.9 | 62.5 $\pm$ 1.4 | 54.3 $\pm$ 0.7 |

---

> > ### Author Response · Authors · 2024-08-14
> > **Kindly Request for Reviewer's Feedback**
> >
> > Dear Reviewer yjRB,
> >
> > We sincerely appreciate the time you have taken to provide feedback on our work, which has helped us greatly improve its clarity, among other attributes. This is a gentle reminder that the Author-Reviewer Discussion period ends at 11:59 pm AoE on August 13. We are happy to answer any further questions you may have before then, but we will be unable to respond after that time.
> >
> > If you agree that our responses to your reviews have addressed the questions you listed, we kindly ask that you consider whether raising your score would more accurately reflect your updated evaluation of our paper. Thank you again for your time and thoughtful comments!
> >
> > Sincerely,
> >
> > Authors

---

> > > ### Comment · Reviewer_yjRB · 2024-08-14
> > > **Response by reviewer**
> > >
> > > Thank the authors for providing further clarification.
> > >
> > > I have read the rebuttal and other reviews, I decide to maintain my original score of 6.

---

> > > > ### Author Response · Authors · 2024-08-14
> > > > **Thank You**
> > > >
> > > > Dear Reviewer yjRB,
> > > >
> > > > Thanks for maintaining positive score of 6 of our paper.
> > > >
> > > > Please let us know if you still have any other questions.
> > > >
> > > > Best,
> > > >
> > > > The Authors

---

### Official Review · Reviewer_rtXt · 2024-07-17

**Soundness:** 3
**Presentation:** 2
**Contribution:** 2
**Rating:** 6
**Confidence:** 4

**Summary:**

The paper proposes FedBNN, a Bayesian approach for personalized federated learning. The approach relies on the availability of an auxiliary small public unlabelled dataset, called Alignment Dataset, that can be used as a mean to distill knowledge across clients. In FedBNN, clients maintain an estimate of a  posterior distribution over the model parameters, which are updated locally using Bayes-by-Backprop. The clients' posteriors are aggregated through Monte Carlo sampling on the  Alignment Dataset. FedBNN has a differentially private implementation which consists in adding a Gaussian noise to the output of the forward pass of the local models on the Alignment Dataset.

The paper conducts thorough numerical experiments to quantify the performance of the FedBNN, and demonstrates that it outperforms SOTA baselines on a wide range of datasets and heterogeneity levels and types.

----
Post Rebuttal

The rebuttal addresses my concerns on the DP guarantees, as such I raise my score to 6.

**Strengths:**

* The paper is overall well written and easy to follow.
* The proposed FedBNN handles both statistical and system heterogeneity. In particular, it allows each client to have a personalized architecture.
* The numerical experiments are rigorous and considers a relatively large number of settings and competitors.

**Weaknesses:**

* FedBNN relies on the availability of a public auxiliary dataset. It is true that an auxiliary dataset is often available, but sometimes it is none.
* I am not sure how to interpret Theorem 4.1 and if the DP mechanism of the paper is correct. On one hand, Theorem 4.1 does not take into consideration the noise multiplier of the DP mechanism, which should be related to $\sigma_g^2$. On the other hand, the DP mechanism presented in the paper does not have any kind of clipping or normalization, which is usually expected. The proof in Appendix C, claims that $\Delta$ is smaller than two, but it only provides a short justification. (*Note: I am not a DP expert*).

**Given the current doubt on Theorem 4.1, I am leaning towards rejection, but I am willing to adjust my rating when this point is clarified.**

**Questions:**

* Could elaborate more on the DP analysis. In particular, I would appreciate if you can formally prove why $\Delta < 2$ in  the proof of Theorem 4.1 in Appendix C. Also, could you please explicitly show the effect of $\sigma_g$ on the DP guarantees in Theorem 4.1.

**Limitations:**

FedBNN relies on the availability of a public auxiliary dataset.

---

> ### Author Rebuttal · Authors · 2024-08-05
>
> Dear Reviewer rtXt,
>
>
> We are thankful to the reviewer for the thorough feedback on our work.
>
> It is encouraging for us to see that the reviewer found the paper easy to read and follow, appreciated the rigorous "*experimental evaluation that considers a wide range of settings and competitors*", and recognized the method's effectiveness in addressing both statistical and system heterogeneity.
>
>
> Below we address the reviewer's concerns on the DP analysis	-
>
> -- *Dependence on $\sigma$*
>
> We acknowledge the doubt regarding not including the noise-parameter $\sigma$ in the main theorem, Theorem 4.1, of the paper. However, we elaborate on this in the proof of the theorem present in the Appendix C (Theorem C.7). The first line in the proof suggests that $\rho = \dfrac{\Delta^2}{2 \sigma^2}$ establishing the relationship between the noise-parameter $\sigma$ and the $(\epsilon, \delta)$ parameters of differential privacy. We have also established this relationship in the Theorem 4.1 in the paper itself in the revised version of the paper.
>
>
>
> -- *Clarification on the bound on $\Delta$*
>
> The sensitivity, denoted as $\Delta$, is defined in Definition C.4 in the paper which defines $L_2$-sensitivity as the maximum change in the $L_2$ norm of the algorithm's output between two neighboring datasets differing in at most one data point. Let $D$ and $D'$ be two neighboring datasets that differ in one data point present at the $i^{th}$ row (without loss of generality), and let $\Phi(D(i,:))$ be the $n_c$ (number of classes) dimensional output probabilities from the model $\Phi$ for the $i^{th}$ row datapoint in $D$ and $\Phi(D'(i,:))$ be the output probabilities for the $i^{th}$ row datapoint in $D'$. The $L_2$ sensitivity of $\Phi$ is -
>
> $\Delta (\Phi) = || \Phi(D) - \Phi(D')||_2$
>
> Since all other data-points between $D$ and $D'$ are identical, the $L_2$ sensitivity of $\Phi$ becomes -
>
> $\Delta (\Phi) = || \Phi(D(i,:)) - \Phi(D'(i,:))||_2$
>
> Now, $\Phi(D(i,:))$ and $\Phi(D'(i,:))$ are both probability distributions, therefore it can be seen that the squared $L_2$ norm of their difference is bounded by 2, i.e., $\Delta(\Phi)^2 \leq 2$ (the maximum occurs when $\Phi(D(i,:))_k = 1$ and $\Phi(D(i,:))_l = 1$ for two separate indices $k \neq l$).
>
> We greatly appreciate the opportunity provided by the reviewer to provide better understanding of the privacy analysis. We have now included these details in the revised version of the paper.
>
> Best,
>
> Authors

---

> ### Author Response · Authors · 2024-08-13
> **Kindly Request for Reviewer's Feedback**
>
> Dear Reviewer rtXt,
>
> We sincerely appreciate the time you have taken to provide feedback on our work, which has helped us greatly improve its clarity, among other attributes. This is a gentle reminder that the Author-Reviewer Discussion period ends in just around 12 hours from this comment, i.e., 11:59 pm AoE on August 13. We are happy to answer any further questions you may have before then, but we will be unable to respond after that time.
>
> If you agree that our responses to your reviews have addressed the questions you listed, we kindly ask that you consider whether raising your score would more accurately reflect your updated evaluation of our paper. Thank you again for your time and thoughtful comments!
>
> Sincerely,
>
> Authors

---

### Decision · Program_Chairs · 2024-09-25

**Decision:**

Accept (poster)

**Comment:**

This paper was a borderline case, but the clarifications and additional experiments provided in the rebuttal helped to lift some concerns, leading to a couple of reviewers increasing their score slightly. Overall, this work introduces a novel and interesting Bayesian approach to personalized FL. I recommend acceptance.

Important note: the authors have committed to add further experiments, as requested by some reviewers. I therefore ask the authors to incorporate these changes in the final version.